# ReDeEP: Detecting Hallucination in Retrieval-Augmented Generation via Mechanistic Interpretability

**Zhongxiang Sun[1], Xiaoxue Zang[2], Kai Zheng[2], Jun Xu[1]\*, Xiao Zhang[1], Weijie Yu [3], Yang Song [2], Han Li [2]**

[1]Gaoling School of Artificial Intelligence, Renmin University of China, Beijing, China
[2]Kuaishou Technology Co., Ltd., Beijing, China
[3]School of Information Technology and Management, University of International Business and Economics
sunzhongxiang@ruc.edu.cn, xxic666@126.com, zhengkai@kuaishou.com, junxu@ruc.edu.cn
zhangx89@ruc.edu.cn, yu@uibe.edu.cn, ys@sonyis.me, lihan08@kuaishou.com

## Abstract

Retrieval-Augmented Generation (RAG) models are designed to incorporate external knowledge, reducing hallucinations caused by insufficient parametric (internal) knowledge. However, even with accurate and relevant retrieved content, RAG models can still produce hallucinations by generating outputs that conflict with the retrieved information. Detecting such hallucinations requires disentangling how Large Language Models (LLMs) utilize external and parametric knowledge. Current detection methods often focus on one of these mechanisms or without decoupling their intertwined effects, making accurate detection difficult. In this paper, we investigate the internal mechanisms behind hallucinations in RAG scenarios. We discover hallucinations occur when the *Knowledge Feedforward Neural Networks* in LLMs overemphasize parametric knowledge in the residual stream, while *Copying Heads* fail to effectively retain or integrate external knowledge from retrieved content. Based on these findings, we propose **ReDeEP**, a novel method that detects hallucinations by decoupling LLM's utilization of external context and parametric knowledge. Our experiments show that ReDeEP significantly improves RAG hallucination detection accuracy. Additionally, we introduce AARF, which mitigates hallucinations by modulating the contributions of Knowledge FFNs and Copying Heads. The source code and dataset are available at https://github.com/Jeryi-Sun/ReDEeP-ICLR.

## 1 Introduction

LLMs have made significant advancements in natural language processing tasks (Dubey et al., 2024; Achiam et al., 2023). However, they still face challenges with hallucinations, often generating factually inaccurate outputs (Huang et al., 2023). To mitigate this issue, many researchers have introduced Retrieval-Augmented Generation (RAG) models, which aim to improve the accuracy of LLM responses by incorporating relevant information retrieved from external knowledge bases (Shuster et al., 2021; Gao et al., 2023).

Despite the use of accurate and relevant retrieved context, RAG models may still produce statements that are either unsupported or contradict the retrieved information, a phenomenon we term **RAG Hallucination** (Niu et al., 2024; Magesh et al., 2024). Recent studies have examined the potential conflicts between the **external context** and the LLM's **parametric knowledge** in RAG models (Xu et al., 2024). As shown in Figure 1, these conflicts can lead to hallucinations but do not always cause them. Therefore, it is important to distinguish RAG hallucination from Knowledge Conflict as a new research direction. Our work focuses on detecting RAG hallucinations, specifically in cases where the retrieved external context is accurate and relevant.

---

*Corresponding author. Work partially done at Engineering Research Center of Next-Generation Intelligent Search and Recommendation, Ministry of Education.

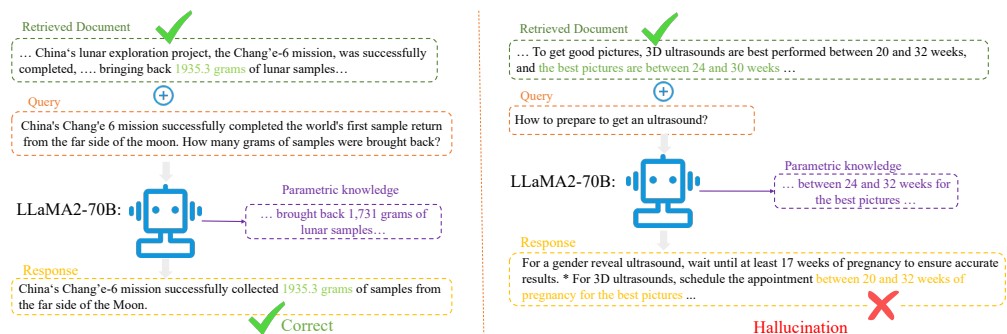

**Figure 1:** Two examples of RAG where the retrieved document is correct but conflicts with parametric knowledge. The left example shows a correct response based on external knowledge, while the right example demonstrates hallucination despite accurate external context.

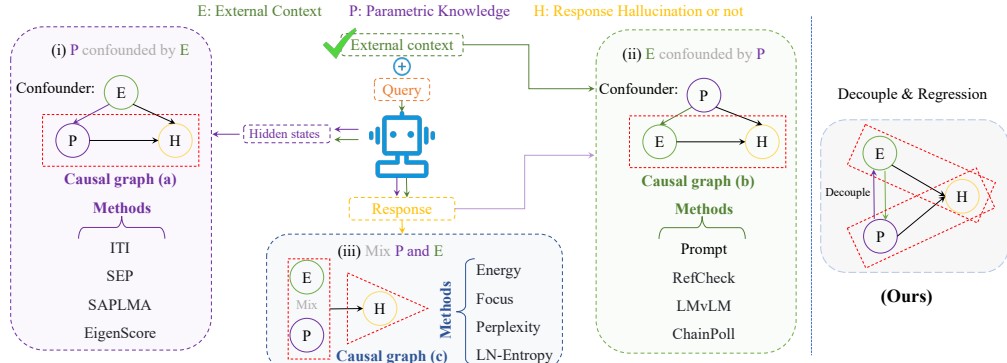

**Figure 2:** Causal perspectives on hallucination detection methods. (i): parametric knowledge is confounded by external context, (ii): external context is confounded by parametric knowledge, and (iii): mixes both without decoupling their contributions. **(Ours)**: decouple these confounders using mechanistic interpretability, incorporating them as covariates to improve hallucination detection.

Existing hallucination detection methods can be categorized into three causal frameworks (Neuberg, 2003; Pearl, 2009), as illustrated in Figure 2 (detailed introduction of methods see Appendix I): (i) **Parametric Confounded by External:** which relies on the LLM's hidden states for hallucination detection, where the external context (E) serves as a confounder between parametric knowledge (P) and hallucinations (H). From a knowledge storage perspective (Geva et al., 2021), hidden states represent the result of querying the parametric knowledge (P) with external context (E), establishing a causal path from E to P (graph (a)). The presence of E as a confounder complicates the accurate prediction of hallucinations based on P alone (Chyzhyk et al., 2022). (ii) **External Confounded by Parametric:** which focuses on hallucination detection by leveraging external context and model responses. Here, parametric knowledge (P) is a confounder between the external context (E) and hallucinations (H), creating a causal link from P to E (graph (b)) due to the unavoidable presence of parametric knowledge in the response. (iii) **Mixed Parametric and External:** which combines both parametric and external knowledge directly, often using uncertainty or sampling techniques (e.g., token probability) to detect hallucinations (graph (c)). However, this mixing of E and P without decoupling their roles obscures their individual contributions (Bengio et al., 2013).

To address the challenges of hallucination detection in RAG models, we first leverage mechanistic interpretability (Ferrando et al., 2024; Elhage et al., 2021) to decouple the LLM's utilization of parametric knowledge and external context. Specifically, we conduct an empirical study to explore the internal mechanisms behind hallucination generation in RAG scenarios. We introduce two metrics: the **External Context Score**, which uses attention heads to quantify the model's utilization on external context, and the **Parametric Knowledge Score**, which is based on FFNs to evaluate LLM's utilization of parametric knowledge (§ 3.1). Correlation analysis and causal intervention reveals that hallucinations typically occur when **Knowledge FFNs** (from later LLM layers) over-add parametric knowledge into the residual stream, while **Copying Heads** (attention heads exhibiting

copying behaviours) neglect the necessary external knowledge from retrieved content or LLM loses the information attended to by Copying Heads during the generation process (§ 3.2).

Building on our causal analysis and mechanistic interpretability, we propose **ReDeEP** (**Re**gressing **De**coupled **E**xternal context score and **P**arametric knowledge score) for detecting hallucinations in LLM-based RAG, which treat parametric knowledge (P) and external context (E) as covariates to solve the confounding problem (Kahlert et al., 2017) (see **Ours** in Figure 2). Additionally, we introduce **AARF** (**A**dd **A**ttention **R**educe **FF**N), which mitigates hallucinations by modulating the contributions of Knowledge FFNs and Copying Heads in the residual stream (§ 4). Experiments on RAGTruth and Dolly (AC) confirm that ReDeEP significantly outperforms existing detection methods, while AARF improves the truthfulness of LLaMA models (§ 5).

## 2 BACKGROUND AND RELATED WORKS

### 2.1 BACKGROUND

Our work is grounded in mechanistic interpretability (Ferrando et al., 2024; nostalgebraist, 2020; Meng et al., 2022; Elhage et al., 2021), which aims to explain how individual components of language models (LMs) contribute to predictions. In this study, we focus on transformer decoder-only architectures (GPT-like models) due to their widespread use (Achiam et al., 2023; Dubey et al., 2024). Transformers use residual connections, where each layer adds information from *Attention Heads* and *Feed-Forward Networks* (FFNs) to the hidden state via the residual stream, contributing to the final prediction (Elhage et al., 2021).

**Attention Heads:** Attention heads play a crucial role in contextualizing token representations by selectively attending to previous tokens and updating the residual stream (Ferrando & Voita, 2024; Clark et al., 2019; Wu et al., 2024). Notably, some attention heads, referred to as *Copying Heads*, have been shown to copy information from one token to another through their OV (output-value) circuits (Elhage et al., 2021). These heads can be identified by analyzing the positive eigenvalues of the OV matrix, which indicate copying behavior. Copying Heads contributes to preserving previously attended tokens in the residual stream, which is critical for external context utilization.

**FFNs:** FFN layers primarily function as knowledge storage in transformers (Geva et al., 2021). Each FFN layer transforms the hidden state by linearly combining key-value pairs, where keys encode specific knowledge and values represent the output of this knowledge. Research shows that FFNs are critical for the utilization of parametric knowledge within LLMs, enabling the model to retrieve and integrate stored information effectively for prediction (Dai et al., 2022).

**Logit Lens:** The LogitLens is a technique that decodes hidden states $x^l$ directly into the vocabulary distribution using the LayerNorm and the unembedding matrix $W_U$ of the LLM for interpretability (nostalgebraist, 2020):

$$\text{LogitLens}\left(x^l\right) = \text{LayerNorm}(x^l)W_U. \tag{1}$$

By understanding the roles of Attention Heads (e.g., Copying Heads) and FFNs, we can better interpret the internal states of LLMs and identify the mechanisms behind hallucinations in RAG scenarios. Detailed background information can be found in Appendix A.

### 2.2 RELATED WORK

**Hallucination of LLMs:** LLMs often generate hallucinations—content inconsistent with real-world facts or inputs (Huang et al., 2023; Sun et al., 2025). As depicted in Figure 2, although there has been extensive research on detecting hallucinations (Niu et al., 2024; Manakul et al., 2023; Han et al., 2024), few studies have concentrated on RAG hallucinations, particularly on the internal mechanisms driving these hallucinations. Inspired by research on Knowledge Conflicts (Xu et al., 2024), our work is the first to apply mechanistic interpretability to lens the internal mechanisms of RAG hallucinations from the perspectives of LLM's utilization of external and parametric knowledge, leading to a more accurate detection method than previous approaches.

**Mechanistic Interpretability:** Mechanistic interpretability (Ferrando et al., 2024; Elhage et al., 2021) seeks to explain the internal processes of LLMs, enabling the interpretation of how individual

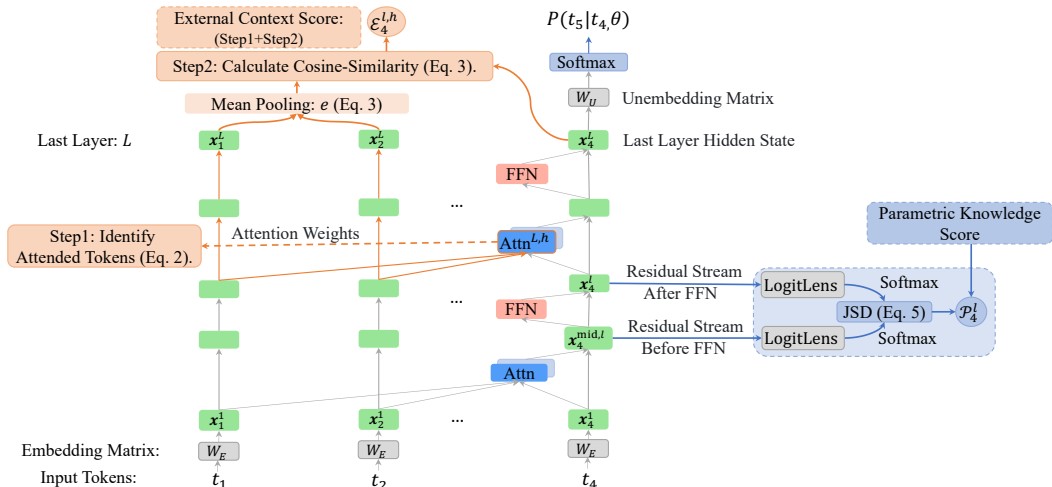

**Figure 3:** Expanded view of Unrolled LLMs' Attention and FFN blocks, illustrating the calculation processes for the External Context Score and Parametric Knowledge Score.

model components contribute to the final prediction. Our work builds on insights into FFN layers, attention heads (Vaswani, 2017), residual streams (Elhage et al., 2021), and the logit lens (nostalgebraist, 2020) to analyze the internal mechanisms of LLMs when RAG hallucinations occur.

## 3 EMPIRICAL STUDY

Our empirical study investigates how hallucinations in RAG models relate to the internal states of the LLM. Using Mechanistic Interpretability techniques, we focus on how the LLM's use of external context and parametric knowledge contributes to hallucinations.

**Experiment Setting:** We conduct experiments on the Llama2-7B-chat model (Touvron et al., 2023) using the training set of RAGTruth dataset (Niu et al., 2024), a high-quality, manually annotated dataset for RAG hallucinations (Details in Section 5.1). Each data point in RAGTruth consists of a query $\mathbf{q}$, retrieved context $\mathbf{c}$, response $\mathbf{r}$, and the hallucination label $h$ (where 0 is truth and 1 is hallucination). During generation, the input to the LLM $f$ is a sequence of tokens $\mathbf{t} = \langle t_1, t_2, \ldots, t_n \rangle$, including the query $\mathbf{q} = \langle t_1, \ldots, t_q \rangle$, retrieved context $\mathbf{c} = \langle t_{q+1}, \ldots, t_c \rangle$, and a partial generated response $\hat{\mathbf{r}} = \langle t_{c+1}, \ldots, t_n \rangle$.

### 3.1 METRICS FOR LLMS' UTILIZATION OF EXTERNAL CONTEXT AND PARAMETRIC KNOWLEDGE

To quantify how LLMs use external context and parametric knowledge, we design two specific metrics, as shown in Figure 3 (a):

**External Context:** Considering attention heads primarily function to retrieve relevant information (as discussed in Section 2.1), we measure the LLM's utilization of external context by assessing (1) whether attention heads focus on the correct context, and (2) whether the LLM effectively retains and utilizes this information during generation. To evaluate these aspects, we define the following metric based on the semantic difference between the external context attended by attention heads and the generated information:

For the last token $t_n$, the attention weights on the context are $\boldsymbol{a}_{n,q:c}^{l,h}$, where $\boldsymbol{a}_n^{l,h}$ is obtained from Equation 8. We select the top $k\%$ tokens with the highest attention scores as attended tokens:

$$\mathcal{I}_n^{l,h} = \arg \text{top}_{k\%}(\boldsymbol{a}_{n,q:c}^{l,h}). \tag{2}$$

Given that attention often shows high sparsity (Zhu et al., 2024; Zhang et al., 2024), with only a few tokens capturing most of the attention scores, we choose $k = 10$ to ensure coverage of high-attention tokens and maintain token diversity.

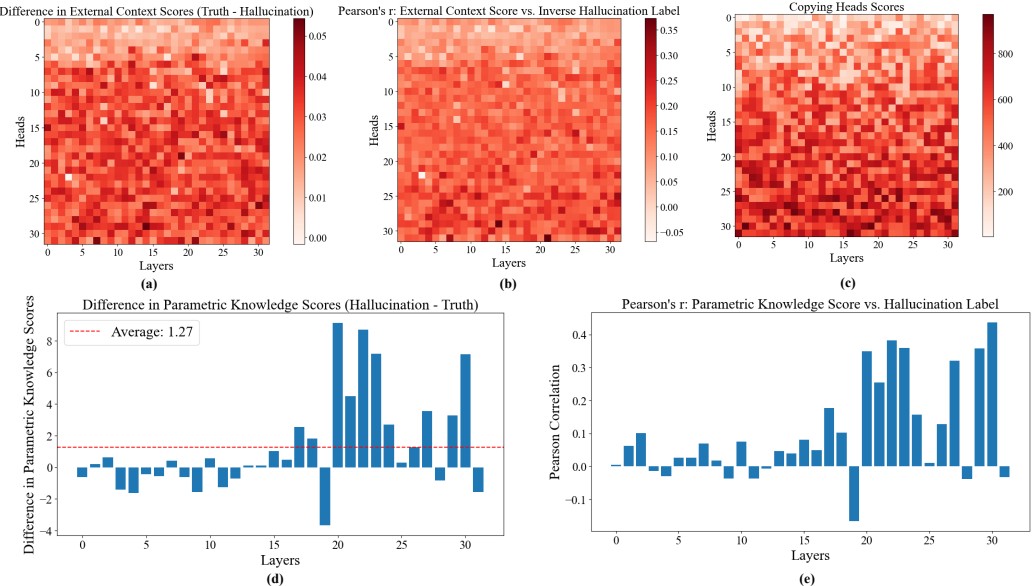

**Figure 4: Relationship Between LLM Utilization of External Context, Parametric Knowledge, and Hallucinations.** Top shows the internal mechanism of LLM's utilization of external context and the occurrence of hallucinations, where the Pearson correlation coefficient between (c) and (a) is 0.41, and between (c) and (b) is 0.46, indicating correlations among them. Bottom illustrates the internal mechanism of LLM's utilization of parametric knowledge and the occurrence of hallucinations, where (d) is scaled by $1e^7$.

Inspired by (Luo et al., 2024; Chen et al., 2024a), which validated that the hidden states of LLM can serve as token semantic representations, we compute the token-level **E**xternal **C**ontext **S**core (ECS) based on the cosine-similarity between the mean-pooling of the last layer hidden states of attended tokens and the hidden state of token $t_n$:

$$\mathcal{E}_n^{l,h} = \frac{\boldsymbol{e} \cdot \boldsymbol{x}_n^L}{\|\boldsymbol{e}\|\|\boldsymbol{x}_n^L\|}, \quad \boldsymbol{e} = \frac{1}{|\mathcal{I}_n^{l,h}|} \sum_{j \in \mathcal{I}_n^{l,h}} \boldsymbol{x}_j^L. \tag{3}$$

The response-level ECS is the average of token-level scores:

$$\mathcal{E}_{\mathbf{r}}^{l,h} = \frac{1}{|\mathbf{r}|} \sum_{t \in \mathbf{r}} \mathcal{E}_t^{l,h}. \tag{4}$$

**Parametric Knowledge:** Considering FFNs store parametric knowledge, to assess how LLM use Parametric Knowledge (as discussed in Section 2.1), we use the LogitLens to map residual stream states before (i.e., $\boldsymbol{x}_n^{\mathrm{mid},l}$, calculated from Equation 9) and after the FFN layer (i.e., $\boldsymbol{x}_n^l$, calculated from Equation 10) to vocabulary distributions. The difference in vocabulary distributions represents the parametric knowledge added by the FFN layer to the residual stream, which is measured by Jensen-Shannon divergence (JSD), gives the token-level **P**arametric **K**nowledge **S**core (PKS):

$$\mathcal{P}_n^l = \mathrm{JSD}\left(q(\boldsymbol{x}_n^{\mathrm{mid},l}) \| q(\boldsymbol{x}_n^l)\right), \tag{5}$$

where $q(\boldsymbol{x}) = \mathrm{softmax}(\mathrm{LogitLens}(\boldsymbol{x}))$. The response-level PKS is the average of token-level scores:

$$\mathcal{P}_{\mathbf{r}}^l = \frac{1}{|\mathbf{r}|} \sum_{t \in \mathbf{r}} \mathcal{P}_t^l. \tag{6}$$

Although these metrics may not be exact due to the complexity of LLMs, they serve as intuitive proxies that are sufficiently aligned with the understanding of existing works to analyze the LLM's use of external context and parametric knowledge in relation to hallucinations, as explored in the following questions.

## 3.2 EXPERIMENTS

**RQ1:** Relationship Between LLM Utilization of External Context, Parametric Knowledge, and Hallucinations

*(1) We first analyze the relationship between **External Context and RAG Hallucinations**:*

**ECS Differences between Truthful and Hallucinated Responses:** To investigate the relationship between LLM utilization of external context and hallucinations, we compare the external context score $\mathcal{E}$ between truthful responses ($h = 0$) and hallucinated responses ($h = 1$). Specifically, we construct two subsets from the dataset: $\mathcal{D}^H$ for hallucinations ($h = 1$) and $\mathcal{D}^T$ for truthful responses ($h = 0$), and calculate the external context score difference for different attention heads:

$$\Delta\mathcal{E}^{l,h} = \mathcal{E}_T^{l,h} - \mathcal{E}_H^{l,h} = \frac{1}{|\mathcal{D}^T|} \sum_{\mathbf{r}\in\mathcal{D}^T} \mathcal{E}_{\mathbf{r}}^{l,h} - \frac{1}{|\mathcal{D}^H|} \sum_{\mathbf{r}\in\mathcal{D}^H} \mathcal{E}_{\mathbf{r}}^{l,h}.$$

**Result:** As shown in Figure 4(a), in Llama2-7B, 1006 out of 1024 attention heads show higher external context scores on the truthful dataset $\mathcal{D}^T$ compared to the hallucination dataset $\mathcal{D}^H$ (i.e., $\Delta\mathcal{E}^{l,h} > 0$). Since the external context score represents the LLM's utilization of external context through attention heads, we can conclude that, at a group level, LLMs utilize external context information less than truthful responses when generating hallucinations.

**Correlation between ECS and Hallucination:** To examine whether neglecting external context relates to RAG hallucinations, we analyzed the Pearson Correlation Coefficient (PCC) between the hallucination label and the external context score across data points in $\mathcal{D}$. Given the expected negative correlation, we inverted the hallucination label $h$ (denoted as $\bar{h}$) and used PCC to quantify the relationship between $\{\bar{h}_i\}_{i=1}^N$ and external context scores $\{\mathcal{E}_i\}_{i=1}^N$.

**Result:** As shown in Figure 4(b), most attention heads show negative correlation between external context scores and hallucination labels $h$. Since the external context score indicates LLMs' utilization of external context, Figure 4(a) and (b) suggest that RAG hallucinations occur when the LLM inadequately leverages external context. Further analysis (Appendix C) shows that hallucinations stem primarily from the LLM losing information attended by attention heads during generation rather than attention heads neglecting external knowledge.

**Relation between Copying Heads and Hallucination:** We observed that the external context score $\mathcal{E}^{l,h}$ of certain attention heads correlates strongly with hallucinations, prompting further exploration of these heads' characteristics. Inspired by the Copying Heads concept from Section 2.1, we examined the relationship between these heads and Copying Heads. The calculation process of each attention head's copying head score $\mathcal{C}^{l,h}$ is shown in Appendix B).

**Result:** As shown in Figure 4(c), the correlation with the results in Figure 4(a) and (b) indicates that attention heads associated with hallucinations are often Copying Heads (PCC between (c) and (a) is 0.41, and (c) and (b) is 0.46). When these Copying Heads have low external context scores, they either fail to attend to the correct external context or, if attended, fail to retain and utilize this information effectively. This reduces the LLM's copying ability and leads to hallucinations, explaining the negative correlation between these heads' external context scores and the hallucination label $h$.

*(2) Next, we analyze the relationship between **Parametric Knowledge and RAG Hallucinations**:*

**PKS Differences between Truth and Hallucination:** We compare the Parametric Knowledge Score $\mathcal{P}$ across different layers when the LLM generates hallucinations versus truthful responses:

$$\Delta\mathcal{P}^l = \mathcal{P}_H^l - \mathcal{P}_T^l = \frac{1}{|\mathcal{D}^H|} \sum_{\mathbf{r}\in\mathcal{D}^H} \mathcal{P}_{\mathbf{r}}^l - \frac{1}{|\mathcal{D}^T|} \sum_{\mathbf{r}\in\mathcal{D}^T} \mathcal{P}_{\mathbf{r}}^l.$$

**Result:** As shown in Figure 4(d), parametric knowledge scores in the later layers of FFN modules are significantly higher in the hallucination dataset compared to the truthful dataset (i.e., $\Delta\mathcal{P}^l > 0$). On average, across all layers, hallucination responses exhibit higher parametric knowledge scores than truthful ones.

**Figure 5:** (Left) Intervention Result for Attention Heads and FFNs. (Right) External Context Scores and Parametric Knowledge Scores (scaled by $1e^5$) comparing Truth & Known (where LLM knows the truthful answer) and Hallucination (where LLM is unknown about the answer and hallucinated).

**Correlation between PKS and Hallucination:** To further explore the relationship between parametric knowledge and hallucinations, we calculate the Pearson correlation between the hallucination label and parametric knowledge scores.

**Result:** As shown in Figure 4(e), parametric knowledge scores in the later layers' FFN modules are positively correlated with the hallucination label $h$ and we define the FFN modules from later layers that show strong correlations with hallucinations as **Knowledge FFNs**. Since these scores represent the amount of parametric knowledge added to the residual stream, we conclude excessive addition of parametric knowledge by these Knowledge FFNs leads to hallucinations. This aligns with findings from LLM early exit studies (Chuang et al., 2024; Schuster et al., 2022): when external context provides sufficient information, shallow layers can generate truthful responses, but over-reliance on parametric knowledge from deeper layers can confuse the model, causing hallucinations.

**RQ2:** Can the relationship identified in RQ1 be validated from a causal perspective?

To validate the causal relationship between Copying Heads, Knowledge FFN modules, and RAG hallucinations identified in Section 3.2, we employ **Causal Intervention** (Ferrando et al., 2024) by intervening on attention heads and FFNs, where we applied noise to the attention scores and amplified the contributions of FFN modules to the residual stream. We compare the Negative Log-Likelihood Loss (NLL) (PyTorch, 2023) difference for the experimental group (Copying Heads/-Knowledge FFNs) and the control group (Other heads/FFNs) on truthful dataset $\mathcal{D}^T$. The detailed intervention procedures are provided in the Appendix E.

**Result:** As shown in Figure 5 (Left), the experimental group's impact on NLL difference was significantly greater than that of the control group for both attention heads and FFN modules. These results, combined with findings from Section 4.3, validate parametric knowledge added by Knowledge FFNs and the ability of Copying Heads to retrieve relevant external knowledge and LLM effectively utilizes this information during generation, have a significant causal relationship with the RAG hallucinations.

> **Finding:** The occurrence of RAG hallucinations is causally related to two primary factors: (1) while the Copying Heads may occasionally neglect necessary knowledge from the external context, a more prominent cause is the LLM losing the Copying Heads retrieved information during the generation process (RQ1-1, RQ2, § C), and (2) the Knowledge FFNs within LLM excessively injecting parametric knowledge into the residual stream (RQ1-2, RQ2, § D).

**RQ3:** Hallucination Behavior Analysis from the Parametric Knowledge Perspective

In this section, we focus on parametric knowledge to analyze hallucination behavior when the LLM either knows or does not know the truthful answer. We conducted a comparison experiment using the LLM-known dataset $\widehat{\mathcal{D}}^T$ and the hallucination dataset $\mathcal{D}^H$ (For detailed analysis, see Appendix F).

**Result:** Our results in Figure 5 (Right) show that when the LLM knows the truthful answer, Copying Heads more accurately capture and utilize external knowledge, and Knowledge FFNs add less parametric knowledge to the residual stream compared to hallucination scenarios, which also supports by (Wadhwa et al., 2024). The results support leveraging our **Finding** to detect RAG hallucination.

## 4 METHODS

Building on our empirical findings (§ 3), we propose **ReDeEP** (Regressing Decoupled External Context and Parametric Knowledge) to detect hallucinations in LLM-based retrieval-augmented generation (§ 4.1, § 4.2), and **AARF** (Add Attention Reduce FFN) to mitigate hallucinations by reweighting the contributions of Knowledge FFNs and Copying Heads to the residual stream (§ 4.3).

### 4.1 TOKEN-LEVEL HALLUCINATION DETECTION — REDEEP (TOKEN)

Our empirical study identified RAG hallucinations as stemming from insufficient utilization of external context by Copying Heads (set $\mathcal{A}$) and excessive reliance on parametric knowledge by Knowledge FFNs (set $\mathcal{F}$). To address the confounding issues shown in Figure 2, we developed a multivariate analysis approach that regresses decoupled External Context Score and Parametric Knowledge Score to predict hallucinations (Kahlert et al., 2017). For a response $\mathbf{r}$, the hallucination score $\mathcal{H}_t$ is:

$$\mathcal{H}_t(\mathbf{r}) = \frac{1}{|\mathbf{r}|} \sum_{t \in \mathbf{r}} \mathcal{H}_t(t), \quad \mathcal{H}_t(t) = \sum_{l \in \mathcal{F}} \alpha \cdot \mathcal{P}_t^l - \sum_{l,h \in \mathcal{A}} \beta \cdot \mathcal{E}_t^{l,h},$$

where $\alpha$ and $\beta$ are regression coefficients for external context and parametric knowledge with $\alpha, \beta > 0$, and this linear regression leverages the high Pearson correlation identified in § 3.

### 4.2 CHUNK-LEVEL HALLUCINATION DETECTION — REDEEP (CHUNK)

As the *Token-level Hallucination Detection* computes scores for each token, it is computationally expensive and lacks full contextual consideration. To improve efficiency and accuracy, we propose *Chunk-level Hallucination Detection* as a more suitable method for RAG hallucination detection. Our approach is inspired by the common chunking operation in RAG Fan et al. (2024); Finardi et al. (2024), where the retrieved context $\mathbf{c}$ and the response $\mathbf{r}$ are divided into manageable segments $\langle \tilde{\boldsymbol{c}}_i \rangle_{i=1}^N$ and $\langle \tilde{\boldsymbol{r}}_j \rangle_{j=1}^M$. For the chunk-level external context score $\hat{\mathcal{E}}^{l,h}$, we first calculate chunk-level attention weights $W_{i,j}^{l,h} = \text{Mean-Pooling}\left(A_{\tilde{\mathbf{c}}_i, \tilde{\mathbf{r}}_j}^{l,h}\right)$, where $A$ is the original token-level attention weight matrix, then determine the highest attention chunk pairs $(\tilde{\mathbf{c}}, \tilde{\mathbf{r}})$. Using an embedding model (emb), we compute the external context score for each chunk as follows:

$$\tilde{\mathcal{E}}_{\mathbf{r}}^{l,h} = \frac{1}{M} \sum_{\tilde{\mathbf{r}} \in \mathbf{r}} \tilde{\mathcal{E}}_{\tilde{\mathbf{r}}}^{l,h}, \quad \tilde{\mathcal{E}}_{\tilde{\mathbf{r}}}^{l,h} = \frac{\text{emb}(\tilde{\mathbf{r}}) \cdot \text{emb}(\tilde{\mathbf{c}})}{\| \text{emb}(\tilde{\mathbf{r}}) \| \| \text{emb}(\tilde{\mathbf{c}}) \|}.$$

For the chunk-level parametric knowledge score $\tilde{\mathcal{P}}^l$, we sum the token-level parametric knowledge scores for each chunk:

$$\tilde{\mathcal{P}}_{\mathbf{r}}^l = \frac{1}{M} \sum_{\tilde{\mathbf{r}} \in \mathbf{r}} \tilde{\mathcal{P}}_{\tilde{\mathbf{r}}}^l, \quad \tilde{\mathcal{P}}_{\tilde{\mathbf{r}}}^l = \frac{1}{|\tilde{\mathbf{r}}|} \sum_{t \in \tilde{\mathbf{r}}} \mathcal{P}_t^l.$$

Finally, the Chunk-level Hallucination Detection score $\mathcal{H}_c(\mathbf{r})$ is defined as:

$$\mathcal{H}_c(\mathbf{r}) = \sum_{l \in \mathcal{F}} \alpha \cdot \tilde{\mathcal{P}}_{\mathbf{r}}^l - \sum_{l,h \in \mathcal{A}} \beta \cdot \tilde{\mathcal{E}}_{\mathbf{r}}^{l,h}.$$

### 4.3 TRUTHFUL RAG GENERATION — AARF

Building on the above methods and the analysis in Appendix C, we propose **Add Attention Reduce FFN (AARF)** to reduce RAG hallucinations by intervening on attention heads and FFN modules without updating model parameters. AARF operates in two stages: (1) token-level hallucination detection and (2) reweighting the contributions of attention heads and FFN modules to the residual stream.

**Table 1:** Performance comparisons between ReDeEP and the baselines. The boldface represents the best performance, and the underline represents the second-best.

| LLMs | Categories | Models | RAGTruth | | | | | Dolly (AC) | | | | |
|---|---|---|---|---|---|---|---|---|---|---|---|---|
| | | | AUC | PCC | Acc. | Rec. | $F_1$ | AUC | PCC | Acc. | Rec. | $F_1$ |
| LLaMA2-7B | MPE | SelfCheckGPT | – | – | 0.5844 | 0.3584 | 0.4642 | – | – | 0.5300 | 0.1897 | 0.3188 |
| | | Perplexity | 0.5091 | -0.0027 | 0.5333 | 0.5190 | 0.6749 | 0.6825 | 0.2728 | 0.6363 | 0.7719 | 0.7097 |
| | | LN-Entropy | 0.5912 | 0.1262 | 0.5600 | 0.5383 | 0.6655 | 0.7001 | 0.2904 | 0.6162 | 0.7368 | 0.6772 |
| | | Energy | 0.5619 | 0.1119 | 0.5088 | 0.5057 | 0.6657 | 0.6074 | 0.2179 | 0.5656 | 0.6316 | 0.6261 |
| | | Focus | 0.6233 | 0.2100 | 0.5533 | 0.5309 | 0.6622 | 0.6783 | 0.3174 | 0.6262 | 0.5593 | 0.6534 |
| | ECP | Prompt | – | – | 0.6700 | 0.7200 | 0.6720 | – | – | 0.6200 | 0.3965 | 0.5476 |
| | | Llama2-13B(LR) | – | – | 0.6350 | 0.7078 | 0.6750 | – | – | 0.6043 | 0.5918 | 0.6616 |
| | | LMvLM | – | – | 0.5946 | 0.7389 | 0.6473 | – | – | 0.6500 | 0.7759 | 0.7200 |
| | | ChainPoll | 0.6738 | 0.3563 | 0.6741 | 0.7832 | 0.7066 | 0.6593 | 0.3502 | 0.6200 | 0.4138 | 0.5581 |
| | | RAGAS | 0.7290 | 0.3865 | 0.6822 | 0.6327 | 0.6667 | 0.6648 | 0.2877 | 0.6500 | 0.5345 | 0.6392 |
| | | Trulens | 0.6510 | 0.1941 | 0.6422 | 0.6814 | 0.6567 | 0.7110 | 0.3198 | 0.6800 | 0.5517 | 0.6667 |
| | | RefCheck | 0.6912 | 0.2098 | 0.6467 | 0.6280 | 0.6736 | 0.6494 | 0.2494 | 0.6100 | 0.3966 | 0.5412 |
| | | P(True) | 0.7093 | 0.2360 | 0.5466 | 0.5194 | 0.5313 | 0.6011 | 0.1987 | 0.5444 | 0.6350 | 0.6509 |
| | PCE | EigenScore | 0.6045 | 0.1559 | 0.5422 | 0.7469 | 0.6682 | 0.6786 | 0.2428 | 0.6596 | 0.7500 | 0.7241 |
| | | SEP | 0.7143 | 0.3355 | 0.6177 | 0.7477 | 0.6627 | 0.6067 | 0.2605 | 0.6060 | 0.6216 | 0.7023 |
| | | SAPLMA | 0.7037 | 0.3188 | 0.5155 | 0.5091 | 0.6726 | 0.5365 | 0.0179 | 0.5600 | 0.5714 | 0.7179 |
| | | ITI | 0.7161 | 0.3932 | 0.5667 | 0.5416 | 0.6745 | 0.5492 | 0.0442 | 0.5800 | 0.5816 | 0.6281 |
| | Ours | **ReDeEP(token)** | 0.7325 | 0.3979 | **0.7067** | 0.6770 | 0.6986 | 0.6884 | 0.3266 | 0.6464 | 0.8070 | 0.7244 |
| | | **ReDeEP(chunk)** | **0.7458** | **0.4203** | 0.6822 | **0.8097** | **0.7190** | **0.7949** | **0.5136** | **0.7373** | **0.8245** | **0.7833** |
| LLaMA2-13B | MPE | SelfCheckGPT | – | – | 0.5844 | 0.3584 | 0.4642 | – | – | 0.5300 | 0.1897 | 0.3188 |
| | | Perplexity | 0.5091 | -0.0027 | 0.5333 | 0.5190 | 0.6749 | 0.6825 | 0.2728 | 0.6363 | 0.7719 | 0.7097 |
| | | LN-Entropy | 0.5912 | 0.1262 | 0.5600 | 0.5383 | 0.6655 | 0.7001 | 0.2904 | 0.6162 | 0.7368 | 0.6772 |
| | | Energy | 0.5619 | 0.1119 | 0.5088 | 0.5057 | 0.6657 | 0.6074 | 0.2179 | 0.5656 | 0.6316 | 0.6261 |
| | | Focus | 0.7888 | 0.4444 | 0.6000 | 0.6173 | 0.6977 | 0.7067 | 0.1643 | 0.5900 | 0.7333 | 0.6168 |
| | ECP | Prompt | – | – | 0.7300 | 0.7000 | 0.6899 | – | – | 0.6700 | 0.4182 | 0.5823 |
| | | Llama2-13B(LR) | – | – | 0.7034 | 0.6839 | 0.7123 | – | – | 0.5545 | 0.6319 | 0.6664 |
| | | LMvLM | – | – | 0.5956 | **0.8357** | 0.6553 | – | – | 0.6300 | 0.7273 | 0.6838 |
| | | ChainPoll | 0.7414 | 0.4820 | 0.7378 | 0.7874 | 0.7342 | 0.7070 | 0.4758 | 0.6800 | 0.4364 | 0.6000 |
| | | RAGAS | 0.7541 | 0.4249 | 0.7000 | 0.6763 | 0.6747 | 0.6412 | 0.2840 | 0.6200 | 0.4182 | 0.5476 |
| | | Trulens | 0.7073 | 0.2791 | 0.6756 | 0.7729 | 0.6867 | 0.6521 | 0.2565 | 0.5700 | 0.3818 | 0.4941 |
| | | RefCheck | 0.7857 | 0.4104 | 0.7200 | 0.6800 | 0.7023 | 0.6626 | 0.2869 | 0.5700 | 0.2545 | 0.3944 |
| | | P(True) | 0.7998 | 0.3493 | 0.6266 | 0.5980 | 0.7032 | 0.6396 | 0.2009 | 0.5600 | 0.6180 | 0.5739 |
| | PCE | EigenScore | 0.6640 | 0.2672 | 0.5267 | 0.6715 | 0.6637 | 0.7214 | 0.2948 | 0.6211 | 0.8181 | 0.7200 |
| | | SEP | 0.8089 | 0.5276 | 0.7288 | 0.6580 | 0.7159 | 0.7098 | 0.2823 | 0.6800 | 0.6545 | 0.6923 |
| | | SAPLMA | 0.8029 | 0.3956 | 0.5488 | 0.5053 | 0.6529 | 0.6053 | 0.2006 | 0.6000 | 0.6000 | 0.6923 |
| | | ITI | 0.8051 | 0.4771 | 0.6177 | 0.5519 | 0.6838 | 0.5511 | 0.0646 | 0.5200 | 0.5385 | 0.6712 |
| | Ours | **ReDeEP(token)** | 0.8181 | 0.5478 | 0.7711 | 0.7440 | 0.7494 | 0.7226 | 0.3776 | 0.6465 | 0.8148 | 0.7154 |
| | | **ReDeEP(chunk)** | **0.8244** | **0.5566** | **0.7889** | 0.7198 | **0.7587** | **0.8420** | **0.5902** | **0.7070** | **0.8518** | **0.7603** |

During the generation of token $t_n$, we compute the hallucination score $\mathcal{H}_t(t_n)$. If $\mathcal{H}_t(t_n) \leq \tau$, we proceed with the normal output computation $f(\mathbf{x})$ (see Equation 12). If $\mathcal{H}_t(t_n) > \tau$, we adjust the weights of Copying Heads $\mathcal{A}$ and Knowledge FFN modules $\mathcal{F}$, shifting focus toward external context and reducing reliance on parametric knowledge:

$$ f(\mathbf{x}) = \sum_{l=1}^{L} \sum_{h=1}^{H} \widehat{\text{Attn}}^{l,h} \left( \boldsymbol{X}_{\leq n}^{l-1} \right) \boldsymbol{W}_U + \sum_{l=1}^{L} \widehat{\text{FFN}}^{l} \left( \boldsymbol{x}_n^{\text{mid},l} \right) \boldsymbol{W}_U + \boldsymbol{x}_n \boldsymbol{W}_U, $$

$$ \widehat{\text{Attn}}^{l,h}(\cdot) = \begin{cases} \alpha_2 \cdot \text{Attn}^{l,h} \left( \boldsymbol{X}_{\leq n}^{l-1} \right), & \text{if } (l,h) \in \mathcal{A}, \\ \text{Attn}^{l,h} \left( \boldsymbol{X}_{\leq n}^{l-1} \right), & \text{otherwise} \end{cases}, \quad \widehat{\text{FFN}}^{l}(\cdot) = \begin{cases} \beta_2 \cdot \text{FFN}^{l} \left( \boldsymbol{x}_n^{\text{mid},l} \right), & \text{if } l \in \mathcal{F}, \\ \text{FFN}^{l} \left( \boldsymbol{x}_n^{\text{mid},l} \right), & \text{otherwise.} \end{cases} $$

Here, $\alpha_2$ is a constant greater than 1 for amplifying attention head contributions, and $\beta_2$ is a constant between (0, 1) for reducing FFN contributions.

## 5 EXPERIMENTS

### 5.1 SETTINGS

**Data:** We evaluate ReDeEP and AARF on two public RAG hallucination datasets. **RAGTruth** is the first high-quality, manually annotated RAG hallucination dataset. The data includes three RAG task

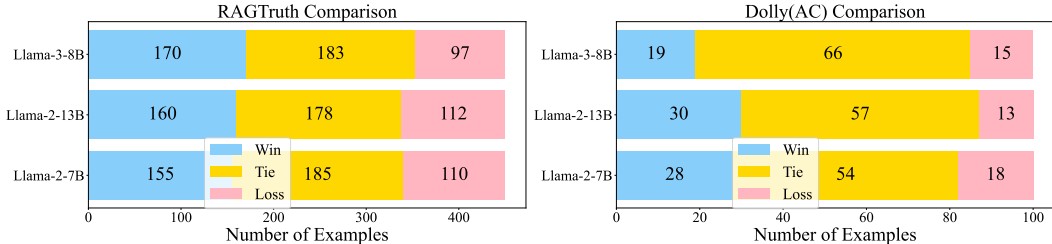

**Figure 6:** Comparison between LLMs+AARF vs LLMs judged by GPT-4o.

types: Question Answering (QA), Data-to-Text Writing, and News Summarization. **Dolly (AC)** is a dataset with Accurate Context obtained from (Hu et al., 2024), including tasks such as text summarization, closed-QA, and information extraction. More details of the data are in Appendix H.

**Baselines:** We conduct experiments on three variants of LLaMA, including LLaMA2-7B-Chat and LLaMA2-13B-Chat. For hallucination detection methods, we follow the classification of existing methods as shown in Figure 2. We use (1) Parametric Confounded by External Methods (**PCE**), (2) External Confounded by Parametric Methods (**ECP**), and (3) Mixed Parametric and External Methods (**MPE**). For detailed baselines information, see Appendix I. We used AUC, Pearson Correlation Coefficient (PCC), Accuracy (Acc.), Recall (Rec.), and $F_1$ as evaluation metrics for detection accuracy. Implementation details are provided in Appendix J.

## 5.2 EXPERIMENTS

**RAG Hallucination Detection:** As shown in Table 1, ReDeEP consistently improves performance across two datasets, various backbone methods, and different metrics, validating its effectiveness in detecting RAG hallucinations. ReDeEP outperforms MPE methods, demonstrating that mechanistic interpretability effectively decouples the LLM's utilization of external context and parametric knowledge, enabling more accurate detection of RAG hallucinations. Additionally, ReDeEP surpasses both ECP and PCE methods by incorporating both the External Context Score and Parametric Knowledge Score as covariates in a multivariate regression approach, effectively addressing the confounding problem. ReDeEP(chunk) generally outperforms ReDeEP(token) in most metrics, suggesting that chunk-level processing better preserves semantic integrity and improves detection performance. Further support for ReDeEP's effectiveness is provided by the ablation study in Appendix K, while the efficiency analysis in Appendix L confirms that ReDeEP achieves comparable time efficiency to the most efficient baselines.

**Truthful RAG Generation:** We evaluated our hallucination reduction method AARF on both the RAGTruth and Dolly (AC) datasets, using GPT-4-o for automatic evaluation to assess the truthfulness (Prompt details can be found in Appendix L). Pairwise comparisons rated by GPT-4-o are shown in Figure 6, demonstrating that AARF can reduce hallucinations to a certain extent compared to the baseline model. These results validate the effectiveness of our intervention experiments and confirm the findings presented in **RQ2** of Section 3.2.

## 6 CONCLUSION

Detecting RAG hallucinations is critical for enhancing the security and reliability of RAG systems. In this work, we introduced ReDeEP, a novel method that detects RAG hallucinations by analyzing LLMs' utilization of parametric knowledge and external context. Our empirical study shows that hallucinations arise from insufficient utilization of external context by Copying Heads and over-reliance on parametric knowledge by Knowledge FFN modules. These insights also guided the development of interventions to reduce hallucinations without updating model parameters. ReDeEP demonstrates significant performance improvements across the LLaMA family and RAG hallucination benchmarks, outperforming existing detection methods.

ACKNOWLEDGMENTS

This work was funded by the National Key R&D Program of China (2023YFA1008704), the National Natural Science Foundation of China (62472426), fund for building world-class universities (disciplines) of Renmin University of China. Work partially done at Engineering Research Center of Next-Generation Intelligent Search and Recommendation, Ministry of Education. Supported by Kuaishou Technology. Supported by the Outstanding Innovative Talents Cultivation Funded Programs 2025 of Renmin University of China.

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

## A  FULL BACKGROUND ON ATTENTION HEADS, FFNS, AND LOGIT LENS

The theoretical foundation of our work is grounded in research on mechanistic interpretability (Ferrando et al., 2024; nostalgebraist, 2020; Meng et al., 2022; Elhage et al., 2021), which seeks to explain the internal processes of language models (LMs) by interpreting how individual model components contribute to the final prediction. In this study, we focus on the transformer decoder-only architecture (also known as GPT-like) due to its widespread success and popularity (Achiam et al., 2023; Dubey et al., 2024). A decoder-only model $f$ consists of $L$ layers and operates on a sequence of embeddings $\mathbf{x} = \langle \boldsymbol{x}_1, \boldsymbol{x}_2, \ldots, \boldsymbol{x}_n \rangle$, which represent the tokens $\mathbf{t} = \langle t_1, t_2, \ldots, t_n \rangle$. Each embedding $\boldsymbol{x} \in \mathbb{R}^d$ is a row vector corresponding to a row of the embedding matrix $\boldsymbol{W}_E \in \mathbb{R}^{|\mathcal{V}| \times d}$, where $\mathcal{V}$ denotes the model's vocabulary. The sequence $\mathbf{x}$ is represented as a matrix $\boldsymbol{X}^0 \in \mathbb{R}^{n \times d}$ with the embeddings stacked as rows.

We interpret Transformers through the perspective of the residual stream (Elhage et al., 2021). Due to the residual connections in Transformers, each layer $l$ takes a hidden state $\boldsymbol{X}^{l-1}$ as input and adds information obtained from its *Attention Heads* and Feed-Forward Networks (*FFNs*) to the hidden state via the residual connection. In this context, the hidden state acts as a residual stream passed through the layers, with each attention and FFN contributing to the final prediction by adding information to the residual stream, resulting in the *Residual Stream States*. The final layer's residual stream state is then projected into the vocabulary space using the *Unembedding Matrix* $\boldsymbol{W}_U \in \mathbb{R}^{d \times |\mathcal{V}|}$ and normalized via the softmax function to produce a probability distribution over the vocabulary, from which a new token is sampled.

The background knowledge for interpreting the contributions of each FFN and attention head to the model's prediction is outlined as follows:

**Attention Heads:** Attention is crucial in Transformers for contextualizing token representations across layers. Each attention head selectively attends to previous positions, gathers information, and updates the current residual stream (Ferrando & Voita, 2024; Clark et al., 2019; Wu et al., 2024). The output of an attention layer is the sum of its attention heads. For each attention head:

$$\text{Attn}^{l,h}\left(\boldsymbol{X}_{\leq i}^{l-1}\right) = \sum_{j \leq i} a_{i,j}^{l,h} \boldsymbol{x}_j^{l-1} \boldsymbol{W}_V^{l,h} \boldsymbol{W}_O^{l,h} = \sum_{j \leq i} a_{i,j}^{l,h} \boldsymbol{x}_j^{l-1} \boldsymbol{W}_{OV}^{l,h} \tag{7}$$

where the learnable weight matrices $\boldsymbol{W}_V^{l,h} \in \mathbb{R}^{d \times d_h}$ and $\boldsymbol{W}_O^{l,h} \in \mathbb{R}^{d_h \times d}$ are combined into the OV matrix $\boldsymbol{W}_{OV}^{l,h} = \boldsymbol{W}_V^{l,h} \boldsymbol{W}_O^{l,h} \in \mathbb{R}^{d \times d}$, also referred to as the OV *(output-value)* circuit (Kobayashi et al., 2021). The attention weights for the current query $i$ to the previous tokens are computed as:

$$\boldsymbol{a}_i^{l,h} = \mathrm{softmax}\left(\frac{\boldsymbol{x}_i^{l-1} \boldsymbol{W}_Q^{l,h} \left(\boldsymbol{X}_{\leq i}^{l-1} \boldsymbol{W}_K^{l,h}\right)^\top}{\sqrt{d_k}}\right) = \mathrm{softmax}\left(\frac{\boldsymbol{x}_i^{l-1} \boldsymbol{W}_{QK}^h \boldsymbol{X}_{\leq i}^{l-1\top}}{\sqrt{d_k}}\right) \quad (8)$$

where $\boldsymbol{W}_Q^{l,h} \in \mathbb{R}^{d \times d_h}$ and $\boldsymbol{W}_K^{l,h} \in \mathbb{R}^{d \times d_h}$ combine to form the QK (query-key) circuit (Elhage et al., 2021), $\boldsymbol{W}_{QK}^h = \boldsymbol{W}_Q^h \boldsymbol{W}_K^{h\top} \in \mathbb{R}^{d \times d}$. The QK circuit computes the attention weights, determining the positions that should be attended, while the OV circuit transfers and transforms the information from the attended positions into the current residual stream. The attention block output is the sum of individual attention heads, which is then added back into the residual stream:

$$\boldsymbol{x}_i^{\mathrm{mid},l} = \boldsymbol{x}_i^{l-1} + \sum_{h=1}^{H} \mathrm{Attn}^{l,h}\left(\boldsymbol{X}_{\leq i}^{l-1}\right) \quad (9)$$

Previous research has shown that the primary role of attention layers in LLMs is implementing algorithms (Olsson et al., 2022; Ferrando et al., 2024). For instance, several attention heads in Transformer LMs have OV matrices exhibiting copying behavior; we refer to these heads as ***Copying Heads***. Elhage et al. (2021) propose using the number of positive real eigenvalues of the full OV circuit matrix $\boldsymbol{W}_E \boldsymbol{W}_{OV} \boldsymbol{W}_U$ as a summary statistic for detecting Copying Heads. Positive eigenvalues indicate that a linear combination of tokens contributes to an increase in the linear combination of logits of the same tokens.

**FFN:** Research has shown that the functionality of FFN layers lies in storing knowledge (Geva et al., 2021). Transformer FFN layers can be represented as a linear combination of vectors. Specifically, for an input vector $\boldsymbol{x}_i^{\mathrm{mid},l} \in \mathbb{R}^d$ drawn from the residual stream states, with FFN parameter matrices $\mathbf{K}^l, \mathbf{V}^l \in \mathbb{R}^{d_m \times d}$, the FFN output can be expressed as:

$$\mathrm{FFN}^l\left(\boldsymbol{x}_i^{\mathrm{mid},l}\right) = g\left(\boldsymbol{x}_i^{\mathrm{mid},l}(\mathbf{K}^l)^T\right) \mathbf{V}^l = \sum_{i=1}^{d_m} f\left(\boldsymbol{x}_i^{\mathrm{mid},l} \cdot \boldsymbol{k}_i^l\right) \boldsymbol{v}_i^l = \sum_{i=1}^{d_m} m_i^l \boldsymbol{v}_i^l$$

$$\boldsymbol{x}_i^l = \boldsymbol{x}_i^{\mathrm{mid},l} + \mathrm{FFN}^l\left(\boldsymbol{x}_i^{\mathrm{mid},l}\right) \quad (10)$$

where $g$ is the activation function. Thus, the FFN layer can be viewed as a linear combination of vectors: the multiplication of $\boldsymbol{x}_i^{\mathrm{mid},l}$ and the key vector $\boldsymbol{k}_i^l$ produces the coefficient $m_i^l$, which weights the corresponding value vector $\boldsymbol{v}_i^l$.

**Logit Lens:** The LogitLens is a technique that decodes hidden states $\boldsymbol{x}^l$ directly into the vocabulary distribution using the LayerNorm and the unembedding matrix of the LLM for interpretability (nostalgebraist, 2020):

$$\mathrm{LogitLens}\left(\boldsymbol{x}^l\right) = \mathrm{LayerNorm}(\boldsymbol{x}^l)\boldsymbol{W}_U \quad (11)$$

This approach has been validated in various studies as an effective method for interpreting LLMs' weight matrices or hidden states (Hanna et al., 2024; Zhou et al., 2024; Yu et al., 2023).

The final output logits of the LLM can be expressed as:

$$f(\mathbf{x}) = \left(\sum_{l=1}^{L}\sum_{h=1}^{H} \mathrm{Attn}^{l,h}\left(\boldsymbol{X}_{\leq n}^{l-1}\right) \boldsymbol{W}_U + \sum_{l=1}^{L} \mathrm{FFN}^l\left(\boldsymbol{x}_n^{\mathrm{mid},l}\right) \boldsymbol{W}_U + \boldsymbol{x}_n \boldsymbol{W}_U\right) \quad (12)$$

## B    Calculation of Copying Heads Score

The traditional method for detecting Copying Heads (Elhage et al., 2021) involves calculating the eigenvalues of the matrix $M = W_U W_{OV}^h W_E$ and assessing the proportion of positive eigenvalues, where $W_E$ is the Embedding Matrix, $W_{OV}$ is the OV Matrix and $W_U$ is the Unembedding Matrix. However, the original Copying Heads identification method proposed by Elhage et al. (2021) requires calculating eigenvalues of large matrices and using the ratio of positive eigenvalues to determine Copying Heads, which becomes computationally expensive for models with large hidden sizes. We propose using the trace of the matrix to estimate the ratio of positive eigenvalues, calibrated with the Gershgorin circle theorem (Bell, 1965), to obtain each attention head's copying head score $\mathcal{C}^{l,h}$. To simplify this, we estimate the proportion using the trace of the matrix and refine this estimation using the Gershgorin Circle Theorem (Bell, 1965).

1. **Trace-Based Estimation:** The trace $\mathrm{tr}(M)$ provides an indication of the distribution of positive and negative eigenvalues:

   - A positive trace suggests more positive than negative eigenvalues.
   - A negative trace suggests more negative than positive eigenvalues.
   - A trace near zero suggests a balance between positive and negative eigenvalues.

2. **Gershgorin Circle Theorem:** To enhance the accuracy of our trace-based estimation, we employ the Gershgorin Circle Theorem, which provides an approximation of the eigenvalue distribution. For any $n \times n$ matrix $M = [a_{ij}]$, each eigenvalue of $M$ lies within at least one Gershgorin disk $D_i$:
$$D_i = \{z \in \mathbb{C} : |z - a_{ii}| \leq R_i\}$$

   where $R_i = \sum_{j \neq i} |a_{ij}|$. Each disk is centered at the diagonal element $a_{ii}$ with a radius determined by the sum of the absolute values of the off-diagonal elements in the row. This theorem helps identify the regions in the complex plane where the eigenvalues are likely to be found, allowing us to approximate their distribution without direct computation.

3. **IQR-Based Outlier Detection for Boundary Points:**

   - Collect boundary points $z + a_{ii}$ and $z - a_{ii}$ from each Gershgorin disk.
   - Calculate the first (Q1) and third quartiles (Q3) of these points, then determine the IQR as $\mathrm{IQR} = Q3 - Q1$ (Vinutha et al., 2018).
   - Identify outliers using bounds $Q1 - 1.5 \times \mathrm{IQR}$ and $Q3 + 1.5 \times \mathrm{IQR}$, counting points outside these limits.

4. **Copying Head Score Calculation:** Combine rankings based on the number of detected outliers (ascending order) and the absolute value of the trace (descending order). Summing these ranks gives the Copying Head Score $\mathcal{C}^{l,h}$, reflecting the head's tendency to behave as a Copying Head.

## C    Dive into the Rationale Behind the External Context Score

In Section 3.1, we designed the external context score to measure two aspects: (1) whether the attention heads focus on the correct external context, and (2) if attending to the correct context, whether the LLM can effectively retain and utilize this information during the generation process. In this section, we aim to explore whether a low external context score is caused by attention heads focusing on the incorrect external context or by the LLM losing the information attended by the attention heads during the generation process.

To address this, we firstly conducted experiments on the Llama2-7B model using the RAGTruth dataset. Specifically, in our validation experiments, we selected data from RAGTruth where LLaMA2-7B-Chat exhibited hallucinations. Using LangChain, a widely-used open-source toolkit, we applied the RecursiveCharacterTextSplitter to segment the input retrieved document into different spans. We then calculated whether the attention module's attended span (by mean pooling the attention scores and selecting the input span with the highest score) during the generation of hallucination spans could identify the hallucination spans in the response. If the attention head successfully

identified the correct span, this indicates that the attention mechanism focused on the correct external context, but the LLM did not effectively retain and utilize this information during the generation process. This evaluation was based on GPT-4-o (from OpenAI) using the following prompt:

> Prompt: {external context + query}
> Respond: {response}
> Conflict Span: {Conflict Span}
> Conflict Type: {Conflict Type}
> Reason: {Reason}
> Given the following context information: {Attend Span}, can this support the existence of a conflict in the response? Please answer with "Yes" or "No" and give the reason on the newline.

**Table 2:** Proportion of data where Llama2-7B attention heads attend to the information that supports conflict.

| Attention heads attend | Attention heads mis-attend |
|---|---|
| 77.5% | 22.5% |

From the results in Table 2, we can see that a low external context score is mostly due to the LLM losing the information attended by the attention heads during the generation process. In most cases, the attention heads correctly attend to the appropriate external context. This phenomena may be due to the presence of some Copy suppression heads in the LLM (McDougall et al., 2023), which may incorrectly suppress the information attended by the Copying head, resulting in the LM losing the information attended by the attention heads during the generation process. This also validates the feasibility of our proposed AARF method in reducing hallucinations by increasing the output of copying heads in the residual stream.

In addition, we aim to verify whether the attention module's attended span during the generation of a hallucinated span contains the information needed in the truthful answer. Our verification data is sourced from RAGTruth, where LLaMA2-7B Chat exhibited hallucinations. For the same questions, we extracted truthful answers provided by other models in the RAGTruth dataset as the ground truth. We filtered out validation data where no truthful answer exists and instances with PKS exceeding the overall average, to minimize the influence of PKS. This evaluation was conducted using GPT-4-o (from OpenAI) with the following prompt:

> Prompt: {external context + query}
> Truthful Respond: {Truthful Respond}
> Given the following context information: {Attend Span}, check if it includes relevant information found in the truthful answer.
> Please answer with "Yes" or "No" and provide the reason on a new line.

**Table 3:** Proportion of data where Llama2-7B attention heads attend to the correct information.

| Attention heads attend truthful information | Attention heads mis-attend truthful information |
|---|---|
| 65.2 % | 34.8% |

The experimental results shown in Table 3 demonstrate that in most cases when hallucinations occur during generation, the attention heads correctly attend to the appropriate external context. This indicates that a low external context score is primarily due to the LLM losing the information attended by the attention heads during the generation process.

**Table 4:** Parametric knowledge bias comparison of Llama2 7B with different layer modifications across datasets."–" denotes a divide-by-zero error during computation.

| Dataset | Llama2 7B | Remove Last (32nd) Layer FFN | Remove 16th Layer FFN | Remove 1st Layer FFN |
|---|---|---|---|---|
| HotpotQA (Yang et al., 2018) | +1.44% | +0.52% ($\Delta$0.92%) | -1.61% ($\Delta$3.05%) | +1.33% ($\Delta$0.11%) |
| SQuAD (Rajpurkar, 2016) | +5.10% | +1.24% ($\Delta$3.85%) | +2.18% ($\Delta$2.92%) | +3.30% ($\Delta$1.80%) |
| NQ (Kwiatkowski et al., 2019) | +4.45% | +2.55% ($\Delta$1.90%) | +2.16% ($\Delta$2.29%) | – |

## D  JUSTIFICATION FOR PARAMETRIC KNOWLEDGE SCORE

To further verify whether the Knowledge FFNs within LLMs excessively injecting parametric knowledge into the residual stream lead to hallucinations, we follow the experimental settings of Table 4 in (Kortukov et al.). Using the Llama2 7B model, we examined whether removing FFN layers reduces parametric knowledge bias. Specifically, in our experiments, we removed the Last (32nd) Layer FFN, the 16th Layer FFN, and the 1st Layer FFN, observing whether the parametric knowledge bias (Kortukov et al.) will decrease. We sampled the first 1,000 examples from the RAG dataset, which is a common setting in RAG scenarios (Jin et al., 2024).

The experimental results are shown in Table 4, where larger values indicate greater parametric knowledge bias. The results demonstrate that removing either lower-layer or mid-to-upper-layer FFNs reduces parametric knowledge bias, with mid-to-upper-layer FFN removal being more effective. This aligns with our finding in the paper that Knowledge FFNs primarily inject parametric knowledge in the mid-to-upper layers. On the HotpotQA dataset, removing the 16th Layer FFN from Llama2 7B even resulted in negative parametric knowledge bias. This further supports our conclusion that, in RAG scenarios equipped with accurate and relevant external knowledge, the Knowledge FFNs within LLMs excessively injecting parametric knowledge into the residual stream can lead to hallucinations.

## E  DETAILED INTERVENTION PROCEDURES

In this section, we provide the details of the intervention experiments for **RQ2** from Section 3.2. Our intervention experiments were conducted on the subset of RAGTruth data which contains truthful answers. The NLL difference was used to measure the change in the negative log-likelihood of the model's output for the truthful answer before and after the intervention. If the NLL difference in the experimental group is larger than that in the control group, it indicates that the intervention in the experimental group is more likely to cause the model to generate non-truthful answers (i.e., hallucinated answers).

**Attention Heads Intervention:** As described in Figure 3 (b), we applied noise to the attention scores $\boldsymbol{a}_i^{l,h}$ of the experimental group to evaluate their impact on hallucinations. Specifically, the attention scores were sampled from a standard normal distribution:

$$a_{i,j}^{l,h} \sim \mathcal{N}(0,1), \quad \tilde{\boldsymbol{a}}_i^{(l,h)} = \text{softmax}\left(\boldsymbol{a}^{(l,h)}\right). \tag{13}$$

This approach simulates the removal of meaningful attention patterns, allowing us to assess how Copying Heads' focus on external context impacts hallucinations.

**FFN Modules Intervention:** To investigate the role of Knowledge FFNs in hallucinations, we amplified the effect of the FFN modules by increasing their contribution to the residual stream tenfold ($k = 10$). This intervention highlights the influence of parametric knowledge on the generation process.

**Causal Matching:** To reduce potential biases arising from the position of transformer layers or heads, we applied causal matching (Stuart, 2010). For attention heads, we matched the top 32 Copying Heads with the nearest non-experimental heads within the same layer. Similarly, we matched the top 5 Knowledge FFNs, identified as being most related to hallucinations, with the nearest FFN modules in adjacent layers. This matching process ensured that the comparison between the experimental and control groups was fair and focused on the specific roles of Copying Heads and Knowledge FFNs in hallucination generation.

## F  DETAILED ANALYSIS OF **RQ3:** HALLUCINATION BEHAVIOR ANALYSIS FROM PARAMETRIC KNOWLEDGE VIEW

In the experiments (**RQ1** and **RQ2** in Section 3.2), we analyzed the relationship between LLM-generated responses and the internal states of the model that lead to hallucinations. In this section, we shift our focus to parametric knowledge (since our setting assumes the external context is correct, there is no need for separate analysis from the external context perspective) to examine the two scenarios where the LLM's internal memory either knows or does not know the truthful answer to the query. This analysis aims to validate whether our previous findings about the connection between LLM hallucinations and internal states are reasonable.

**LLM Parametric Knowledge Unknown Truthful Answer** When the LLM's parametric knowledge does not contain the truthful response, the model must rely on the retrieved context to generate a truthful answer. In this scenario, the Knowledge FFN module may over-add parametric knowledge to the residual stream, while Copying Heads may fail to attend to the correct external context or lose the attended external information during the generation process, leading to hallucinations. This phenomenon is consistent with our earlier findings, where Copying Heads neglect external context and Knowledge FFN modules excessively add parametric knowledge to the residual stream.

**LLM Parametric Knowledge Known Truthful Answer** When the LLM's parametric knowledge contains the truthful answer, RAG responses are typically truthful. To verify whether the model, in this case, relies more on external knowledge and less on parametric knowledge compared to when hallucinations occur, we designed a validation experiment. Specifically, we allowed the LLM to generate responses directly on the truthful dataset $\mathcal{D}^T$ without relying on retrieved documents to determine if the LLM could independently produce accurate answers. We used GPT-4-o (Achiam et al., 2023), along with the original truthful response, to evaluate whether the LLM-generated answers matched the expected ones, thus assessing if the LLM's parametric knowledge could correctly answer independently (see prompt in Appendix G). These correct responses form the LLM-known dataset $\widehat{\mathcal{D}}^T$. We then analyzed the differences in external context scores and parametric knowledge scores for the LLM across $\widehat{\mathcal{D}}^T$ and $\mathcal{D}^H$.

**Result:** As shown in Figure 5 (Right), when the LLM's parametric knowledge knows the truthful answer, we observe that Copying Heads can more accurately capture external knowledge and effectively retain and utilize this information during the generation process, showing more stable performance compared to their behavior in the hallucination dataset. Although in scenarios where the LLM knows the truthful answer, the Knowledge FFN layers add less parametric knowledge to the residual stream than the hallucination dataset, this supports our earlier finding of the negative impact of excessive utilization of parametric knowledge by the Knowledge FFN module.

## G  PROMPT FOR EVALUATING PARAMETRIC KNOWLEDGE

> You are an AI evaluator tasked with assessing the accuracy and relevance of an AI-generated response. Here are the details:
>
> 1. AI-generated response: {*LLM-Generated-Response*}
>
> 2. Expected response: {*Ground Truth*}
>
> 3. Query that prompted the response: {*Query*}
>
> Evaluate if the AI-generated response accurately and comprehensively addresses the query and aligns with the expected response. If the AI-generated response aligns well with the expected response, output "yes". If it does not align, output "no". Only output "yes" or "no".

To assess whether the LLM's parametric knowledge alone could provide accurate answers independently of retrieved documents, we used the following prompt in our validation experiment. The aim was to evaluate if the LLM-generated responses on the truthful dataset $\mathcal{D}^T$ matched the expected truthful answers. The prompt was designed to engage GPT-4-o (Achiam et al., 2023) as an evaluator to compare the LLM's responses with the ground truth.

## H    DETAILS ABOUT RAG HALLUCINATION DATASETS

**RAGTruth:** The RAGTruth (Niu et al., 2024) dataset is the first high-quality, manually annotated RAG hallucination dataset. It is divided into three task types: Question Answering (QA), Data-to-Text Writing, and News Summarization. However, the dataset does not include hallucination annotations for responses generated by Llama3-8B. To address this, we employed three annotators with graduate-level qualifications to manually evaluate the presence of hallucinations in different LLM RAG responses. Each response was carefully assessed to determine whether it contained any hallucinations based on the accuracy and relevance of the retrieved and generated content.

**Dolly (AC):** The Dolly (AC) dataset is sourced from (Hu et al., 2024) and consists of tasks such as text summarization, closed-QA, and information extraction. Similar to RAGTruth, Dolly (AC) lacks hallucination annotations for responses generated by certain LLMs, particularly those without access to accurate context. To fill this gap, the same team of three qualified annotators was tasked with manually evaluating the RAG responses to determine if they contained hallucinations, focusing on the alignment between the generated responses and the external context.

In both cases, the manual annotation process involved cross-verifying the generated content with the provided external context to detect discrepancies or factual inconsistencies that would indicate hallucinations. We consider some data that fails to gain consensus as low-quality data which can be filtered out.

## I    DETAILS ABOUT BASELINE MODELS

This section provides details on the baseline models and hallucination detection methods used in our experiments. We categorize the methods into three groups based on their approach to leveraging parametric knowledge and external context: Parametric Confounded by External (PCE), External Confounded by Parametric (ECP), and Mixed Parametric and External (MPE).

**(1) Parametric Confounded by External (PCE):**

- **EigenScore:** EigenScore measures the semantic consistency in the embedding space. Higher EigenScores suggest a higher likelihood of hallucinations, as they indicate greater semantic divergence (Chen et al., 2024b).

- **SEP:** SEP (Semantic Entropy Probe) uses linear probes trained on the hidden states of LLMs to detect hallucinations by analyzing the semantic entropy of the tokens before generation (Han et al., 2024).

- **SAPLMA:** SAPLMA trains a classifier on LLM activation values to detect hallucinations. It captures internal signals from the LLM's hidden layers to identify when the model might generate a hallucination (Azaria & Mitchell, 2023).

- **ITI:** Inference-Time Intervention (ITI) analyzes attention head activations and uses a binary classifier to predict hallucinations by studying the relationship between heads and task performance (Li et al., 2024).

**(2) External Confounded by Parametric (ECP):**

- **Prompt:** This method uses the prompts provided in the RAGTruth (Niu et al., 2024) to evaluate whether the LLM-generated responses are hallucinations by comparing them against the ground truth using GPT-4-o-mini (Niu et al., 2024).

- **Llama2-13B(LR)**: Fine-tune the Llama2-13B model using LoRA (Hu et al.)  with the training data from RAGTruth for hallucination detection. The LoRA parameter settings are based on the paper (Zheng et al., 2024).

- **LMvLM:** LMvLM employs a multi-turn interaction between two language models(GPT-4-o-mini vs. Backbone LLM) to discover inconsistencies by having them cross-examine each other's responses (Cohen et al., 2023).

- **ChainPoll:** ChainPoll uses GPT-4-o-mini to determine if a completion contains hallucinations through a carefully designed prompt. The evaluation is repeated multiple times

(typically five), and the final hallucination score is calculated as the ratio of "yes" answers to the total number of responses (Friel & Sanyal, 2023).

- **RAGAS:** RAGAS checks the faithfulness of the generated response by breaking down sentences into shorter assertions and verifying each against the context, using GPT-4-o-mini to calculate a faithfulness score as the ratio of supported statements (Es et al., 2024).

- **Trulens:** Trulens assesses the overlap of information between the context and the generated response using GPT-4-o-mini, assigning a groundedness score between 0 and 10 based on the degree of overlap (Trulens, 2024).

- **RefCheck:** Similar to RAGAS, RefCheck extracts knowledge graphs from the generated responses and evaluates whether the knowledge graphs align with the external context (Hu et al., 2024).

- **P(True):** P(True) measures the uncertainty of the generated claim by querying the LLM itself on the truthfulness of its generated response. The confidence score is calculated as the probability of the first token being "True" (Kadavath et al., 2022).

**(3) Mixed Parametric and External (MPE):**

- **SelfCheckGPT:** SelfCheckGPT uses a zero-resource, sampling-based approach where multiple reference responses are checked by GPT-4-o-mini for consistency with the generated answer (Manakul et al., 2023).

- **LN-Entropy:** Length-Normalized Entropy measures sequence-level uncertainty across multiple generations, using entropy normalized by sequence length to detect hallucinations (Malinin & Gales, 2020).

- **Energy:** Energy-based OOD detection identifies hallucinations by analyzing the uncertainty in the generated response using energy functions. Higher energy suggests a higher likelihood of hallucinations (Liu et al., 2020).

- **Focus:** Focus enhances uncertainty-based hallucination detection by focusing on key informative tokens, preceding words, and token properties, simulating human factuality checking (Zhang et al., 2023).

- **Perplexity:** This method uses the perplexity of the LLM-generated response to detect hallucinations. A higher perplexity indicates greater uncertainty and a higher likelihood of hallucinations (Ren et al., 2022).

## J  IMPLEMENTATION DETAILS.

We run all the experiments on machines equipped with NVIDIA V100 GPUs and 52-core Intel(R) Xeon(R) Gold 6230R CPUs at 2.10GHz. We utilize the Huggingface Transformers package to conduct experiments. During the decoding of responses from the language models, we employ greedy search to generate responses. The remaining parameters follow the models' default settings. For RAGTruth, we use the validation set to select the hyperparameters. For Dolly (AC), we use two-fold validation to select the hyperparameters. For the baselines, we perform hyperparameter tuning within the range provided by the original works. For experiments with randomness, we run them three times to calculate the average. For ReDeEP(Chunk) on Dolly (AC), on Llama2-7B, we select the top-7 scoring Copying Head and top-3 FFN layers with $\alpha = 1$ and $\beta = 1.6$, as described in Section 3. On Llama2-13B, we select the top-11 scoring Copying Head and top-3 FFN layers with $\alpha = 1$ and $\beta = 0.2$. On Llama3-8B, we select the top-1 scoring Copying Head and top-1 FFN layers with $\alpha = 1$ and $\beta = 0.1$, as described in Section 3. For ReDeEP(Chunk) on RAGTruth, on Llama2-7B, we select the top-3 scoring Copying Head and top-4 FFN layers with $\alpha = 1$ and $\beta = 0.6$. On Llama2-13B, we select the top-9 scoring Copying Head and top-3 FFN layers with $\alpha = 1$ and $\beta = 1.8$. On Llama3-8B, we select the top-2 scoring Copying Head and top-5 FFN layers with $\alpha = 1$ and $\beta = 1.2$. For ReDeEP(Token) on Dolly (AC), on Llama2-7B, we select the top-4 scoring Copying Head and top-3 FFN layers with $\alpha = 1$ and $\beta = 0.2$, as described in Section 3. On Llama2-13B, we select the top-4 scoring Copying Head and top-5 FFN layers with $\alpha = 1$ and $\beta = 0.6$. On Llama3-8B, we select the top-1 scoring Copying Head and top-1 FFN layers with $\alpha = 1$ and $\beta = 0.1$, as described in Section 3. For ReDeEP(Token) on RAGTruth, on Llama2-7B, we select

the top-1 scoring Copying Head and top-10 FFN layers with $\alpha = 1$ and $\beta = 0.2$. On Llama2-13B, we select the top-2 scoring Copying Head and top-17 FFN layers with $\alpha = 1$ and $\beta = 0.6$. On Llama3-8B, we select the top-3 scoring Copying Head and top-30 FFN layers with $\alpha = 1$ and $\beta = 0.4$. For AARF on RAGTruth, for Llama2-7B and Llama2-13B, we select $\alpha_1 = 5$, $\beta_1 = 0.2$, and $\tau = 0.4$, while for Llama3-8B, we use $\alpha_1 = 5$, $\beta_1 = 0.2$, and $\tau = 0.0$. On Dolly (AC), for Llama2-7B, Llama2-13B, and Llama3-8B, we select $\alpha_1 = 2$, $\beta_1 = 0.5$, and $\tau = 0.6$. As proposed in Section 4.2, ReDeEP (Chunk) requires segmenting the retrieved documents and responses from the benchmark. For this, we utilized LangChain [1], a popular open-source toolkit, and applied the RecursiveCharacterTextSplitter for the segmentation process. We use BGE embeddings Xiao et al. (2023) for ReDeEP(chunk), which is the Sota embedding model. The thresholds for the prediction of ReDeEP(token) and ReDeEP(chunk) are shown in Table 5.

For the top $K$ for attention heads ($\mathcal{A}$) and top $K$ for FFNs ($\mathcal{F}$), we performed an initial ranking of the correlations between the ECS of different attention heads and the PKS of FFN layers with hallucinations on the validation set. Then, we employed a greedy grid search method. For the top-$K$ attention heads and top-$K$ FFN layers, $K$ was searched within the range of $[1, 32]$. For the remaining two parameters, we used two approaches: the first was to treat them as trainable parameters and regress on the validation set, and the second, adopted in this paper, was to directly perform a grid search. Specifically, we first fixed one parameter, $\alpha$, to 1, and then tuned $\beta$. We found that $\beta$ could be adjusted within the range of $(0, 2)$ with a step size of 0.1. For text chunking, we utilized LangChain's `RecursivelySplitByCharacter`, which automatically splits text based on characters. Since this is not the focus of our research, we set the maximum chunk size to the commonly used 256 in LangChain, with a chunk overlap of 20. For AARF, the parameters are relatively simple. We used grid search for $\tau$ in the range of $(0, 1)$ with a step size of 0.1. For the weight $\alpha_2$, grid search was performed within the values $[1, 2, 5, 10]$, and for $\beta_2$, grid search was performed in the range of $(0, 1)$ with a step size of 0.1.

The Llama2-7B can be downloaded from `https://huggingface.co/meta-llama/Llama-2-7b-chat-hf`. The Llama2-13B can be downloaded from `https://huggingface.co/meta-llama/Llama-2-13b-chat-hf`. The Llama3-8B can be downloaded from `https://huggingface.co/meta-llama/Meta-Llama-3-8B-Instruct`. The BGE embeddings can be downloaded from `https://huggingface.co/BAAI/bge-base-en-v1.5`.

**Table 5:** Threshold of ReDeEP(Token) and ReDeEP(Chunk) across different models and dataset.

|  | RAGTruth | | Dolly | |
| --- | --- | --- | --- | --- |
|  | ReDeEP(Token) | ReDeEP(Chunk) | ReDeEP(Token) | ReDeEP(Chunk) |
| LLaMA2-7B | 0.6 | 0.3 | 0.4 | 0.3 |
| LLaMA2-13B | 0.6 | 0.6 | 0.6 | 0.2 |
| LLaMA3-8B | 0.4 | 0.4 | 0.2 | 0.2 |

## K  ABLATION STUDY

As shown in Table 6, when performing RAG hallucination detection, using only the Parametric Knowledge Score (Only PKS) or only the External Context Score (Only ECS) does not achieve the same performance as the Full ReDeEP model. This validates the effectiveness of employing multivariate regression, where both PKS and ECS are used simultaneously as covariates. According to the analysis in Section 1, using Only PKS or Only ECS introduces confounding issues, leading to a decrease in performance. This explains why both Only ECS and Only PKS yield lower results compared to Full ReDeEP.

## L  EFFICIENCY ANALYSIS

As shown in Figure 7, ReDeEP (chunk) is more efficient than ReDeEP (token), confirming the superior time efficiency of our proposed chunk-level hallucination detection method. Additionally,

---

[1] `https://www.langchain.com/`

**Table 6:** Ablation Study of ReDeEP.

| **RAGTruth** | | | | | | | |
| --- | --- | --- | --- | --- | --- | --- | --- |
| **ReDeEP (Token)** | | AUC | PCC | **ReDeEP (Chunk)** | | AUC | PCC |
| **LLaMA2-7B** | Only PKS | 0.6950 | 0.3327 | **LLaMA2-7B** | Only PKS | 0.6180 | 0.2103 |
| | Only ECS | 0.7234 | 0.3779 | | Only ECS | 0.7098 | 0.3944 |
| | Full | 0.7325 | 0.3979 | | Full | 0.7458 | 0.4203 |
| **LLaMA2-13B** | Only PKS | 0.7214 | 0.3682 | **LLaMA2-13B** | Only PKS | 0.6614 | 0.2566 |
| | Only ECS | 0.8040 | 0.5201 | | Only ECS | 0.7231 | 0.3922 |
| | Full | 0.8181 | 0.5478 | | Full | 0.8244 | 0.5566 |
| **LLaMA3-8B** | Only PKS | 0.6102 | 0.1085 | **LLaMA3-8B** | Only PKS | 0.6082 | 0.1695 |
| | Only ECS | 0.7336 | 0.4312 | | Only ECS | 0.6923 | 0.3102 |
| | Full | 0.7522 | 0.4493 | | Full | 0.7285 | 0.3964 |
| **Dolly (AC)** | | | | | | | |
| **ReDeEP (Token)** | | AUC | PCC | **ReDeEP (Chunk)** | | AUC | PCC |
| **LLaMA2-7B** | Only PKS | 0.6671 | 0.2374 | **LLaMA2-7B** | Only PKS | 0.6383 | 0.2115 |
| | Only ECS | 0.6629 | 0.2852 | | Only ECS | 0.7552 | 0.4478 |
| | Full | 0.6884 | 0.3266 | | Full | 0.7949 | 0.5136 |
| **LLaMA2-13B** | Only PKS | 0.6639 | 0.2891 | **LLaMA2-13B** | Only PKS | 0.6790 | 0.2883 |
| | Only ECS | 0.6856 | 0.3107 | | Only ECS | 0.7383 | 0.4338 |
| | Full | 0.7226 | 0.3776 | | Full | 0.8420 | 0.5902 |
| **LLaMA3-8B** | Only PKS | 0.6329 | 0.2300 | **LLaMA3-8B** | Only PKS | 0.7334 | 0.3503 |
| | Only ECS | 0.6105 | 0.1556 | | Only ECS | 0.6166 | 0.2624 |
| | Full | 0.6701 | 0.2421 | | Full | 0.7354 | 0.3652 |

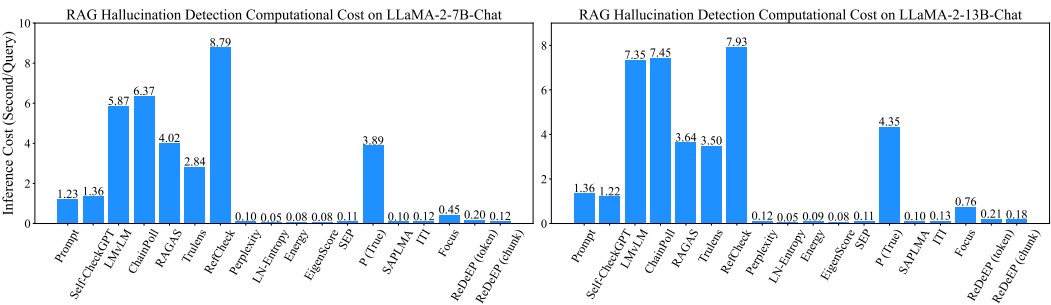

**Figure 7:** Comparison of the detection efficiency of different hallucination detection methods on LLaMA-2-7B-Chat and LLaMA-2-13B-Chat.

methods that do not rely on external models (e.g., ITI, ReDeEP, etc.) exhibit significantly higher efficiency compared to those using external models (e.g., RefCheck, Chainpoll, etc.). ReDeEP is positioned at a high-efficiency level among hallucination detection models, validating its feasibility for industrial applications. Considering that the main time consumption of AARF lies in ReDeEP (token), the relatively low time consumption of ReDeEP (token) further validates the efficiency of AARF.

## M    PROMPT FOR TRUTHFULNESS EVALUATION

In the Truthful RAG Generation Experiments, we used the following anonymous prompt to assess the truthfulness of RAG generation from AARF (A) and Baseline (B) models:

You are tasked with evaluating two answers generated by different models: A and B. Your goal is to determine which answer is more truthful based on the provided content and query. Please carefully compare the truthfulness of both answers. Follow the specific response format provided below.
**Query:** {*String*}
**External Context:** {*String*}
**A answer:** {*String*}
**B answer:** {*String*}
Respond in the following format:
{ {
"win_model": "A (A's answer is more truthful) or B (B's answer is more truthful) or Tie (Both answers are equally truthful. Generally not needed. Choose this only if no distinction can be made)",
"reason": "Provide the reason for selecting the winning model's answer as more truthful."
}}

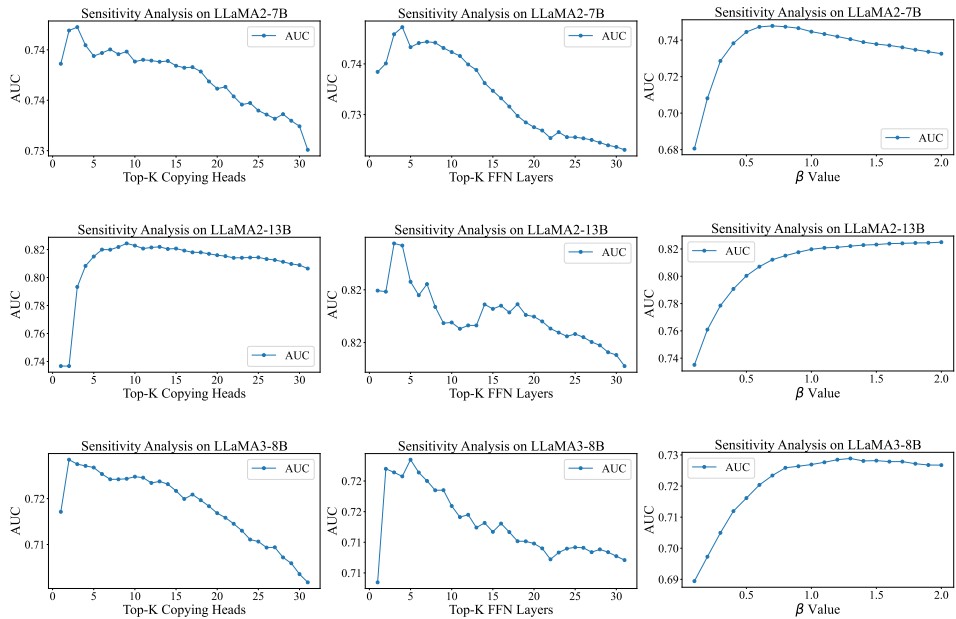

**Figure 8:** Sensitivity Analysis of ReDeEP(chunk).

# N  SENSITIVITY ANALYSIS

In this section, we conduct sensitivity analysis experiments for ReDeEP(Token) and ReDeEP(Chunk) based on the parameter ranges provided in Section J. From the experimental results in Figure 8 and Figure 9, we observe that ReDeEP(Token) and ReDeEP(Chunk) exhibit an initial increase followed by a decrease or a stable trend for most parameters. The limited performance variation range indicates that individual parameters have minimal impact on the overall performance, demonstrating the stability of our model. Furthermore, the sensitivity curves for the same hyperparameters across different models are similar. The performance extrema for most hyperparameters appear within the same parameter range across different models, further showcasing the cross-model stability of ReDeEP.

# O  CASE STUDY FOR AARF

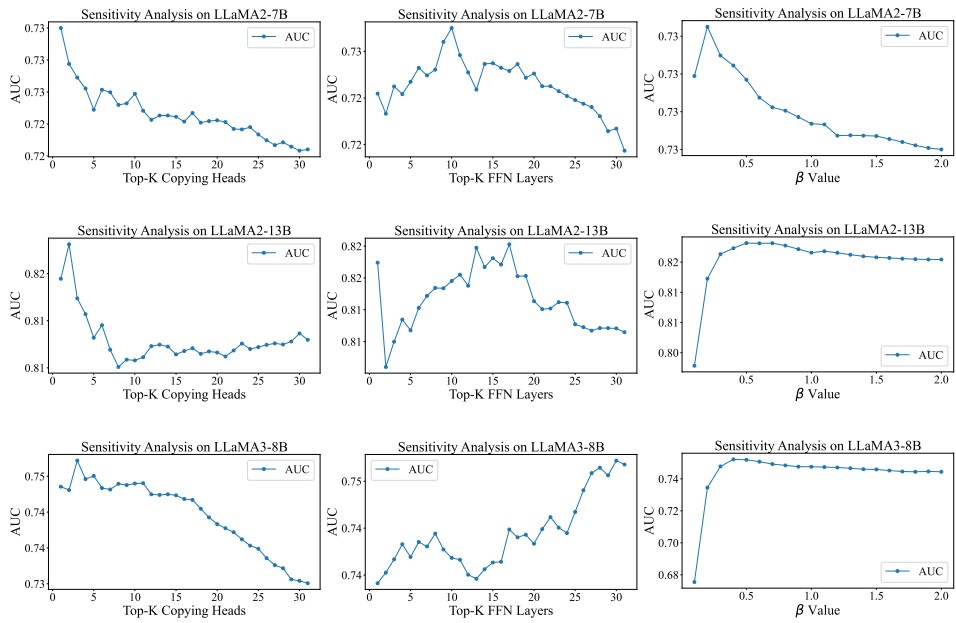

**Figure 9:** Sensitivity Analysis of ReDeEP(token).

In this section, we analyze the results of a case study for AARF on LLaMA2-7B RAGTruth data. From the examples where A is the AARF and B is the Baseline, we observe that AARF can enhance the comprehensiveness of the generation (Case One), more accurately attribute referenced content to the correct passage (Case Two), and reduce fabricated hallucinations (Case Three), all without affecting coherence or writing quality. These results highlight the advantages of AARF in mitigating RAG hallucinations.

---

**Example One: Part One**

**Instruction:** Write an objective overview about the following local business based only on the provided structured data in the JSON format. You should include details and cover the information mentioned in the customers' review. The overview should be 100 - 200 words. Don't make up information. Structured data:
{'name': "Sam's To Go", 'address': '315 Meigs Rd, Ste I', 'city': 'Santa Barbara', 'state': 'CA', 'categories': 'Sandwiches, Restaurants, American (New)', 'hours': {'Monday': '10:0-21:0', 'Tuesday': '10:0-21:0', 'Wednesday': '10:0-21:0', 'Thursday': '10:0-21:0', 'Friday': '10:0-21:0', 'Saturday': '10:0-21:0', 'Sunday': '10:0-19:30'}, 'attributes': {'BusinessParking': {'garage': False, 'street': False, 'validated': False, 'lot': True, 'valet': False}, 'RestaurantsReservations': False, 'OutdoorSeating': True, 'WiFi': 'free', 'RestaurantsTakeOut': True, 'RestaurantsGoodForGroups': True, 'Music': None, 'Ambience': {'touristy': False, 'hipster': False, 'romantic': False, 'divey': False, 'intimate': False, 'trendy': False, 'upscale': False, 'classy': False, 'casual': False}}, 'business_stars': 4.5, 'review_info': [{'review_stars': 5.0, 'review_date': '2022-01-03 00:05:08', 'review_text': "This was our first time at Sams and it was delicious!! Excellent customer service, they were so sweet. We got the #11 Italian cold cuts, and the veggie sandwich #29 both were so good! The bread was so fresh and you could tell the ingredients were really quality! My daughter got the turkey sandwich and she is the pickiest eater and she loved it! Can't wait to go back"},

**Example One: Part Two**

{'review_stars': 5.0, 'review_date': '2021-12-10 03:34:00', 'review_text': "The couple who runs the place are so nice and funny, and the sandwiches are delicious. It doesn't look fancy, but do yourself a favor and try it out. The Pastrami Avocado and Cheese was delicious. Warning, the medium size sandwich is BIG."}, {'review_stars': 2.0, 'review_date': '2021-11-28 21:43:45', 'review_text': 'Iv́e been going to Samś for the past several years while visiting beautiful Santa Barbara from the San Francisco Bay area. Iv́e always enjoyed the sandwiches from Samś. Unfortunately, my wife and I just finished eating their turkey sandwich and in all honesty, itẃas just, Ok, more like disappointing... We were disappointed because it appears the quality and taste of the sandwich has digressed to a "point of no return"... The sandwich was sloppily made, poorly wrapped, and importantly, the man behind the counter (usual guy) seems to have gotten cantankerous over time. He chastised me for not ordering both white and wheat bread sandwiches the way he wanted, getting perturbed for asking for olives in my sandwich and, he had a smart alec - bad attitude demeanor about him - poor customer service.

Suffice it to say, we will not be going back to Samś. There are too many good delis in the Cliffs Dr. area to be subjected to poor quality sandwiches and an unpleasant customer experience...'}]} Overview:

**A's Response:** Sam's To Go is a popular sandwich shop located in Santa Barbara, California. With a rating of 4.5 stars, the business offers a variety of delicious sandwiches, including Italian cold cuts, veggie sandwiches, and more. The atmosphere is described as casual and welcoming, with free WiFi and outdoor seating available. Customers have praised the excellent customer service and high-quality ingredients, with one reviewer noting that the bread is always fresh and of high quality. However, one customer had a negative experience with the service, finding the staff to be unfriendly and the sandwich to be poorly made. Overall, Sam's To Go is a great option for sandwich lovers looking for a quick and satisfying meal in a cozy setting.

**B's Response:** Sam's To Go is a popular sandwich shop located in Santa Barbara, California. With a rating of 4.5 stars, the business offers a variety of delicious sandwiches, including Italian cold cuts, veggie sandwiches, and more. The atmosphere is casual and welcoming, with outdoor seating available. Customers have praised the excellent customer service and high-quality ingredients used in their sandwiches. One reviewer noted that the sandwiches are made fresh and with quality ingredients, while another appreciated the friendly and funny staff. However, one reviewer had a disappointing experience with their turkey sandwich, finding it to be poorly made and lacking in taste. Despite this, Sam's To Go remains a popular choice for sandwich lovers in the area.

**GPT4-o Judge Result:** Model A's answer provides a more comprehensive overview of both positive and negative customer experiences while maintaining the core information from the structured data. It accurately describes the atmosphere, the quality of the sandwiches, and includes specific feedback about customer service, which reflects the nuances in the reviews better than Model B. Model B lacks some details on the specific issues raised in the negative review, making A's response more truthful and balanced.

---

**Example Two**

**Instruction:** Briefly answer the following question: how to boil potatoes for easy peeling Bear in mind that your response should be strictly based on the following three passages: passage 1:Boil until tender, then transfer to an ice water bath. When cool, peel off skin. (15 minutes to cook, 5 seconds to peel each cooked potato) And now, an epic demonstration of How to Peel 2 Pounds of Potatoes in Under A Minute using a drill, (clean) toilet brush, bucket, and water hose. GAME. CHANGER.

passage 2:This versatile medium-starch spud has a delicious, buttery flavor. It cooks up tender and holds its shape, but it also has just enough starch to contribute creaminess without making a potato salad mushy. Its skin can be tough and papery, so we recommend peeling them first.

passage 3:Especially if you are peeling a crap load of potatoes. Here is the easiest way (or at least the easiest way I've discovered) to peel a cooked potato. Step 1.) Bring a pot of water to a boil. Step 2.) With a knife, score your potatoes all the way around the center. Step 3.) Place your potatoes in the pot and boil for about 15 minutes or so. Step 4.) Prepare a bowl of ice water and place your potatoes in the water. Step 5.) Swish it around for a few seconds just enough so that it's cool enough to handle. Step 6.) Remove your potatoe from the water and remove the skin.

**A' Response:** Based on the provided passages, the best way to boil potatoes for easy peeling is to:
1. Boil them until tender, then transfer them to an ice water bath to cool down quickly (as shown in passage 1). 2. Peel the potatoes as soon as they are cooked, as the skin can become tough and papery if left to sit (as stated in passage 2). 3. Use a drill, toilet brush, bucket, and water hose to peel 2 pounds of potatoes in under a minute (as demonstrated in passage 3).

**B's Response**: Based on the provided passages, the best way to boil potatoes for easy peeling is to:
1. Boil them until tender, then transfer them to an ice water bath to cool down quickly (as shown in passage 1). 2. Peel the potatoes as soon as they are cooked, as the skin can become tough and papery if left to sit (as mentioned in passage 2). 3. Boiling them, scoring the skin, cooling them in ice water, and then effortlessly removing the skin. (as demonstrated in passage 3).

**GPT4-o Judge Result:** A's answer accurately summarizes and organizes the critical steps for boiling potatoes for easy peeling, referencing the instructions from the passages sequentially. It correctly includes the process of boiling, cooling in an ice water bath, and peeling.

---

**Example Three: Part One**

**Instruction:** Summarize the following news within 193 words: We want our killers to be kind. To walk into court and show remorse, tell us that they are really nice people who only did it because they feared for their lives, or they were temporarily insane. Even though those things may not be true, too often they are accepted as legitimate excuses for murder. Well, Aaron Hernandez, the former New England Patriots tight end and now convicted murderer, made none of those excuses. He walked into court with an air of bravado, his head held high like the $40 million, NFL superstar he was just a couple of years ago. Several times he was even caught winking at his fiancee, Shayanna Jenkins, during the trial. And we didn't like his swagger. Hernandez didn't offer a plausible alibi. He didn't look ashamed or remorseful. He never wept. His own attorney, James Sultan, admitted that Hernandez "witnessed" the killing of Odin Lloyd, "committed by somebody he knew," but said his client did not commit the crime. Even before the guilty verdict came down Wednesday, for many – at least those covering the trial – audacity seemed to be the defendant's biggest crime. "The Arrogance of Aaron Hernandez," a New Yorker headline accused. I don't get it. What does it matter that Hernandez was arrogant in court, or walked with too much swagger, or even smiled at his girl? Like it or not, that is who he is. The evidence is what matters. And for once, this time it appears the jury carefully considered the damning mountain of circumstantial evidence against this defendant and came to the right decision: guilty of first-degree murder in the 2013 slaying of Odin Lloyd. Hernandez, 25, was sentenced to life without the possibility of parole. He was also found guilty of unlawful possession of a firearm and unlawful possession of ammunition. It was the right decision. Still, it's hard not to feel sadness over such a senseless waste of life for both Lloyd and now Hernandez, who were once friends. Football was the best thing Hernandez ever had, especially after he father died suddenly after a routine hernia surgery in 2006. Hernandez was 16. After that his life got complicated. At 17, Hernandez went off to the University of Florida and seemed headed for greatness. There he won the John Mackey Award as the nation's best tight end and led the team in receiving during its 2009 Bowl Championship Series win. But off the field, his life was to beginning to unravel. Trouble started: bar fights, reports of marijuana use and failed drug tests. He was even questioned in relation to a shooting after a fight at a Gainesville, Florida, nightclub. By the time he was drafted in 2010 by the New England Patriots, Hernandez had already been labeled a "troubled player." But neither the Patriots nor the NFL has anything to be ashamed of in how they handled the Hernandez case. They did everything right this time. Hours after Hernandez was charged with murder, he was let go from the team. And his coaches didn't spend any time trying to convince the public that their star tight end was a decent, family-loving guy, as the league has done too many times in the past when its players got in trouble. This time justice worked. There's no reason to second-guess what went wrong, to ask "How could a star NFLer be a murderer?" The NFL for the most part does a great job vetting its players, and certainly Hernandez is an anomaly in the league, where despite the ugly headlines, the overwhelming majority of players are upstanding, law-abiding citizens. And Patriots bashers (me included) would be mistaken to try to find fault with the team for drafting Hernandez despite his troubled past. No one really knows what evil lurks in the hearts of others, even those closest to us. Not the mothers whose sons go off and commit schoolyard killing sprees, or wives whose husbands gun down innocent people, and certainly not employers who are mostly concerned about performance on the job. There was no way to predict Hernandez would end up a murderer. He was a guy who had all the talent and opportunity in the world, but he still went wrong. Hernandez himself may have explained it best as he was being taken out of the courtroom:. According to a law enforcement source close to the case, Hernandez told officers escorting him, "'Hey man, I'm going to miss you guys. ... I don't need any luck any more.' He gave you the impression, 'It's kinda like no big deal. ... It is what it is.' "
output:

---

---

**Example Three: Part Two**

**A's Response:** Aaron Hernandez, a former New England Patriots tight end, was convicted of first-degree murder and sentenced to life in prison without the possibility of parole. Despite his arrogant demeanor in court, the jury found him guilty of killing Odin Lloyd in 2013. Hernandez showed no remorse and offered no excuses for his actions, and his own attorney admitted that he witnessed the killing but did not commit it. The case highlights the problem of excusing violent behavior by athletes and celebrities, and the need for more thorough vetting of NFL players. The NFL did the right thing by releasing Hernandez from the team hours after he was charged, and there is no reason to second-guess their decision. The case is a reminder that evil can hide in plain sight, and that even those with the best of opportunities can still go wrong.

**B's Response:** Aaron Hernandez, a former New England Patriots tight end, was convicted of first-degree murder and sentenced to life in prison without parole. Despite his $40 million NFL career and seemingly charmed life, Hernandez showed no remorse in court and was found guilty of killing Odin Lloyd in 2013. The jury deliberated for less than a day before delivering the guilty verdict. Hernandez's defense team argued that he was not the killer, but the evidence against him was overwhelming. The case has raised questions about the NFL's vetting process and how it handles players with troubled pasts. Many are left feeling saddened by the senseless waste of life, as both Lloyd and Hernandez had promising futures that were cut short.

**GPT4-o Judge Result:** Model A's answer focuses on the key elements of the case, including Hernandez's conviction, sentencing, and lack of remorse, while also addressing the concerns about the societal tendency to excuse violent behavior and the NFL's response. It accurately captures the overall context of the situation, emphasizing the responsibility of the judicial process and the moral implications of Hernandez's actions, which aligns more closely with the original content's message.

