# OpenReview forum: "ReDeEP: Detecting Hallucination in Retrieval-Augmented Generation via Mechanistic Interpretability"
_ICLR.cc/2025/Conference — ICLR 2025 Spotlight_

### Official Review · Reviewer_ixxW · 2024-11-03

**Soundness:** 3
**Presentation:** 3
**Contribution:** 3
**Rating:** 6
**Confidence:** 3

**Summary:**

This work proposes ReDeEP - a method to detect hallucinations by LLMs in retrieval augmented generation settings by using mechanistic interpretability. The authors introduce two novel metrics - (1) the External Context Score (ECS) and (2) Parametric Knowledge Score (PKS) to identify when hallucinations happen because of over reliance on internal knowledge or from the underuse of external information. The authors also introduce AARF (Add Attention Reduce FFN), which aims to adjust the weights of the attention heads, and feed forward layers to reduce hallucinations.  Their approach is empirically validated on standard benchmarks, demonstrating superior performance to existing methods.

**Strengths:**

1. The development of the ECS and PKS metrics to understand the contributions external and internal knowledge have on the LLM's generation is a compelling and novel way to understand LLM outputs.

2. They demonstrated great empirical validation by running extensive experiments across two datasets, three LLMs, and many baseline methods.

3. They also introduce a method to curb hallucinations called AARF - which relates back to the introduced metrics nicely.

**Weaknesses:**

1. Performing this analysis at the token/chunk level might limit its practicality in real time or large scale settings - it would be nice to have a richer discussion of the trade-offs and real world feasibility.

2. The experiments are extensive - however they are all with the LLama family of models - testing (even a much smaller set) on a different model would be informative.

3. While the performance of AARF seems good (Figure 6) - it would be good to see some example outputs - its unclear how this could effect the model’s output in terms of coherence/writing in general.

**Questions:**

1. Could you discuss more the trade-offs of your method? In particular thinking about real time settings?

2. Have you tested your method on non-LLama models? Do you anticipate any challenges for different models?

3. Could you provide some example outputs pre and post using AARF? Or can you speak to the effect AARF has on the coherence of the model’s output after AARF?

---

> ### Author Response · Authors · 2024-11-20
>
> **Q1: Performing this analysis at the token/chunk level might limit its practicality in real time or large scale settings - it would be nice to have a richer discussion of the trade-offs and real world feasibility.**
>
> **A1:** We appreciate the your concern about the practicality of token/chunk-level analysis in real-time or large-scale applications. However, ReDeEP is designed to calculate PKS and ECS metrics in parallel during response generation, ensuring low latency. As detailed in Appendix L Line 1232, our efficiency analysis shows that ReDeEP achieves competitive performance among hallucination detection methods, validating its industrial feasibility. While ReDeEP achieves high efficiency and low latency, we acknowledge that additional optimizations, such as selective chunk-level analysis or adaptive granularity, could further enhance its applicability in extremely resource-constrained or large-scale scenarios.
>
> ---
>
>
> **Q2: The experiments are extensive - however they are all with the LLama family of models - testing (even a much smaller set) on a different model would be informative.**
>
> **A2:** Thank you for your question. To address your concern, we have added experiments on the Mistral-7B-Instruct model using the RAGTruth dataset. Due to time constraints, we selected representative baselines and compared them against ReDeEP (Chunk). The results demonstrate that ReDeEP (Chunk) outperforms baseline models on most metrics, showing that ReDeEP can generalize beyond the LLama family to effectively detect hallucinations.
>
> | Model          | AUC    | PCC    | ACC    | Precision | Recall  | F1     |
> |-----------------|--------|--------|--------|-----------|---------|--------|
> | LMvLM          | \      | \      | 0.5977 | 0.6023    | **0.8207**  | 0.6948 |
> | Prompt         |  \     | \      | 0.6933 | 0.7070    | 0.7689  | 0.7366 |
> | RAGA           | 0.6748 | 0.2048 | 0.6844 | 0.6835    | 0.8088  | 0.7409 |
> | Chainpoll      | 0.7064 | 0.4228 | 0.7178 | 0.7214    | 0.8048  | 0.7608 |
> | Refcheck       | 0.7060 | 0.1475 | 0.6667 | 0.7265    | 0.6454  | 0.6835 |
> | Trulens        | 0.6836 | 0.1476 | 0.6511 | 0.7136    | 0.6255  | 0.6667 |
> | ReDeEP (Chunk) | **0.7257** | **0.4269** | **0.7332** | **0.7865**    | 0.7945  | **0.7854** |
>
> These results highlight the capability of ReDeEP to detect hallucinations effectively on models outside of the LLama family, thereby validating its generalizability.

---

> ### Author Response · Authors · 2024-11-20
>
> ---
>
>
> **Q3: While the performance of AARF seems good (Figure 6) - it would be good to see some example outputs - its unclear how this could effect the model’s output in terms of coherence/writing in general.**
>
> **A3:** Thank you for your suggestion. In the revised paper, we have added a new section, Appendix O: CASE STUDY FOR AARF (Line 1349), which analyzes the results of AARF on the LLaMA2-7B RAGTruth dataset. Due to the length of the examples, we only present one example here (A is the AARF and B is the Baseline):
>
> ```
> Example Two:
> Instruction: Briefly answer the following question:
> how to boil potatoes for easy peeling
> Bear in mind that your response should be strictly based on the following three passages:
> passage 1:Boil until tender, then transfer to an ice water bath. When cool, peel off skin. (15 minutes to cook, 5 seconds to peel each cooked potato) And now, an epic demonstration of How to Peel 2 Pounds of Potatoes in Under A Minute using a drill, (clean) toilet brush, bucket, and water hose. GAME. CHANGER.
>
> passage 2:This versatile medium-starch spud has a delicious, buttery flavor. It cooks up tender and holds its shape, but it also has just enough starch to contribute creaminess without making a potato salad mushy. Its skin can be tough and papery, so we recommend peeling them first.
>
> passage 3:Especially if you are peeling a crap load of potatoes. Here is the easiest way (or at least the easiest way I’ve discovered) to peel a cooked potato. Step 1.) Bring a pot of water to a boil. Step 2.) With a knife, score your potatoes all the way around the center. Step 3.) Place your potatoes in the pot and boil for about 15 minutes or so. Step 4.) Prepare a bowl of ice water and place your potatoes in the water. Step 5.) Swish it around for a few seconds just enough so that it’s cool enough to handle. Step 6.) Remove your potatoe from the water and remove the skin.
>
>
> A' Response: Based on the provided passages, the best way to boil potatoes for easy peeling is to:
>
> 1. Boil them until tender, then transfer them to an ice water bath to cool down quickly (as shown in passage 1).
> 2. Peel the potatoes as soon as they are cooked, as the skin can become tough and papery if left to sit (as stated in passage 2).
> 3. Boiling them, scoring the skin, cooling them in ice water, and then effortlessly removing the skin. (as demonstrated in passage 3).
>
>
> B's Response: Based on the provided passages, the best way to boil potatoes for easy peeling is to:
>
> 1. Boil them until tender, then transfer them to an ice water bath to cool down quickly (as shown in passage 1).
> 2. Peel the potatoes as soon as they are cooked, as the skin can become tough and papery if left to sit (as mentioned in passage 2).
> 3. Use a drill, toilet brush, bucket, and water hose to peel 2 pounds of potatoes in under a minute (as demonstrated in passage 1).
>
>
> GPT4-o Judge Result: A's answer accurately summarizes and organizes the critical steps for boiling potatoes for easy peeling, referencing the instructions from the passages sequentially. It correctly includes the process of boiling, cooling in an ice water bath, and peeling.
> ```
>
> From the results of the examples presented in Appendix O, we observe that AARF enables the model to:
> - Generate more comprehensive responses (*Case One*),
> - Attribute cited content more accurately to the correct passage (*Case Two*), and
> - Reduce fabricated hallucinations (*Case Three*),
>
> without negatively affecting coherence or writing quality. These findings demonstrate the advantages of AARF in reducing hallucinations in RAG scenarios.

---

> > ### Comment · Reviewer_ixxW · 2024-11-24
> > **Reviewer Response**
> >
> > Thank you for your detailed response.
> >
> > I feel all my questions were addressed and I would like to keep my review positive.

---

> > > ### Author Response · Authors · 2024-11-25
> > >
> > > Thank you for your positive feedback and support. We are glad all your questions were addressed!

---

### Official Review · Reviewer_JEtc · 2024-11-03

**Soundness:** 4
**Presentation:** 3
**Contribution:** 4
**Rating:** 8
**Confidence:** 4

**Summary:**

Retrieval-Augmented Generation (RAG) models are still prone to hallucinations. This paper explores the internal mechanisms behind hallucinations in RAG settings. Building on these insights, the authors propose a hallucination detection method, ReDeEP, and a RAG truthfulness improvement method, AARF.

**Strengths:**

Each step is thoughtfully motivated, with both conceptual reasoning and empirical validations in §3. The detection method shows effective results in Table 1, and the RAG truthfulness improves using AARF, as shown in Figure 6.

**Weaknesses:**

Figure 3 is problematic. The starting point and flow of the diagram are unclear, with too many arrows, making it hard to identify the main computational paths. An effective graphic would show one main data processing pipeline, which is missing here. Additionally, the quantities computed are not well-defined. Panels (b) and (c) add no extra information and could be removed without loss.

Otherwise, rather minor points:
- l.281: Please describe the number of hallucinations and non-hallucinations (h = 0 and h = 1) in the evaluation set.
- Pearson's Correlation in §3: Why measure Pearson’s correlation between ECS and hallucination labels (binary)? It would be more informative to report accuracy at a fixed threshold or detection metrics such as AUROC. Similarly, for PKS and hallucination, detection metrics like AUROC would be preferable.
- l.465: Could you clarify the criteria for selecting thresholds for accuracy, recall, and F1?

Even more nits:
- Use full names for FFN, ReDeEP, and AARF, at least in the abstract.
- In Figure 4(c), clarify what the colour bar values represent.
- Overall, font sizes in the figures are too small.
- Structure in §3.2 is difficult to follow. Stick to a standard structure using \section, \subsection, \subsubsection, \paragraph, etc., rather than introducing new hierarchies (boldface, underline, italics, numbering (1), (2), …).

**Questions:**

The paper offers valuable insights into how RAG-based LLMs produce hallucinated outputs. Building on these findings, it proposes a detection method and mitigation strategy grounded in this understanding. Presentation issues remain, particularly in the main figure explaining the method, yet the contribution is significant.

---

> ### Author Response · Authors · 2024-11-20
>
> **Q1: Figure 3 is problematic.**
>
> **A1:**
> Thank you very much for your insightful suggestions. We have carefully revised Figure 3 based on your feedback. Specifically:11_
>
> 1. To reduce ambiguity and improve clarity, we removed non-essential components such as the FFN and attention components for tokens $t_1$ and $t_2$, retaining only the main residual stream.
> 2. To simplify the diagram and reduce visual clutter, we removed token $t_3$, which also reduced the number of arrows.
> 3. We added more detailed descriptions for input and output variables, isolating the computation graphs for the Parametric Knowledge Score and External Context Score to better define their computation processes.
> 4. Panels (b) and (c) have been removed to focus solely on the main model diagram.
> 5. The font size has been increased to enhance readability.
>
> The updated version of Figure 3 can be found in our revised paper.
>
> ---
>
> **Q2: The number of hallucinations and non-hallucinations (h = 0 and h = 1) in the evaluation set**
>
> **A2:** Thank you for your question. The number of hallucinations (h = 1) and non-hallucinations (h = 0) in the evaluation set are summarized below:
>
> | Model          | Dataset       | Hallucinations:Non-hallucinations |
> |----------------|---------------|------------------------------------|
> | LLaMA2-7B      | RAGTruth      | 226:224                           |
> |                | Dolly (AC)    | 58:42                             |
> | LLaMA2-13B     | RAGTruth      | 207:243                           |
> |                | Dolly (AC)    | 55:45                             |
> | LLaMA3-8B      | RAGTruth      | 243:207                           |
> |                | Dolly (AC)    | 41:59                             |
>
>
> ---
>
> **Q3: Question about Pearson's Correlation in §3.**
>
> **A3:** Thank you for your question. The purpose of using Pearson's correlation in Section 3 is to study whether there exists a statistically significant linear relationship between the External Context Score (ECS)/Parametric Knowledge Score(PKS) and hallucination labels. This analysis is aimed at validating our hypothesis about the relationship between the two variables, rather than assessing detection performance. While detection metrics such as AUROC and accuracy provide insight into the classification performance, Pearson's correlation serves as a measure of the statistical relationship between ECS/PKS and hallucination labels. In the experimental section, we have already included AUROC and other detection metrics to assess the classification performance of using ECS and PKS in detecting hallucinations. The use of Pearson's correlation in Section 3 is limited to an exploratory analysis to identify potential linear trends between the variables, rather than as an evaluation metric for the detection task.
>
> **Q4: Criteria for selecting thresholds for accuracy, recall, and F1**
>
> **A4:** Thank you for your question. We have summarized the criteria for selecting thresholds for accuracy, recall, and F1 in the following table:
>
> | Model        | RAGTruth (token) | RAGTruth (chunk) | ReDeEP (token) | ReDeEP (chunk) |
> |--------------|------------------|------------------|----------------|----------------|
> | LLaMA2-7B    | 0.6              | 0.3              | 0.4            | 0.3            |
> | LLaMA2-13B   | 0.6              | 0.6              | 0.6            | 0.2            |
> | LLaMA3-8B    | 0.4              | 0.4              | 0.2            | 0.2            |
>
> ---
>
> **Q5: Use full names for FFN, ReDeEP, and AARF, at least in the abstract.**
>
> **A5:** Thank you for your question. We have updated the abstract to replace "FFN" with its full name. Additionally, we have included explanatory modifiers for "ReDeEP" and "AARF" in the abstract to ensure better clarity and understanding for readers.
>
> ---
>
> **Q6: In Figure 4(c), clarify what the colour bar values represent.**
>
> **A6:** Thank you for your suggestion. In Figure 4(c), the color bar values represent the copying head scores. We have updated the title of Figure 4(c) in the revised paper to clarify this.
>
> ---
>
> **Q7: Structure in §3.2 is difficult to follow. Stick to a standard structure using \section, \subsection, \subsubsection, \paragraph, etc.**
>
> **A7:** Thank you very much for your suggestion. The structure in §3.2 was organized this way primarily to save space, considering the strict page limitations for the main body of ICLR submissions. We will adopt a standard structure in other versions of the paper to make it easier for readers to follow.

---

> > ### Comment · Reviewer_JEtc · 2024-11-25
> >
> > Looks good, keeping my score. Thanks for the great effort answering questions :)

---

> > > ### Author Response · Authors · 2024-11-25
> > >
> > > Thank you very much for your response and positive feedback!

---

### Official Review · Reviewer_LdfE · 2024-11-04

**Soundness:** 3
**Presentation:** 2
**Contribution:** 3
**Rating:** 8
**Confidence:** 4

**Summary:**

This paper proposes a method for detecting hallucinations of Retrieval Augmented Generation (RAG) models in the scenario when retrieved context is accurate and relevant.

The authors hypothesize that hallucinations are caused by models ignoring the retrieved context and overemphasizing their parametric knowledge. To capture these concepts they introduce two auxilary scores: External Context Score (ECS) that reflects utilization of the retrieved context by the model, and Parametric Knowledge Score (PKS) that reflects utilization of the parametric knowledge. Hallucinations are then predicted by thresholding a hallucination score H which is computed as a weighted sum of ECS and PKS.

In addition to that, the authors propose a method to reduce hallucinations by suppressing outputs of attention heads that contribute to low ECS and outputs of fully-connected layers that contribute to high PKS.

**Strengths:**

- Authors provide a straightforward method to detect hallucinations in RAGs that does not require model fine-tuning.
- Empirical results provided by the authors look good.

**Weaknesses:**

## Lack of justification for PKS and ECS

### No PKS justification
Although PKS is correlated with a hallucination label (line 319) there is still no guarantee that it is adding parametric knowledge. Since you do not provide any theoretical justification for this score, at least an empirical justification is needed. You can run a simple experiment: use LogitLensed outputs before FFN as final outputs and check whether it removes the parametric knowledge bias using some of the setups for that, for example, the one from [1] (they study it through the prism of the phenomenon you encounter in RQ3 and Appendix E).

### Questionable ECS justification
Contrary to the PKS the authors provided empirical justification for the ECS measuring model reliance on context, however, I find it not very convincing so far.

First of all, I do not see how the ratio of attention head attending vs mis-attending justifies ECS. It would make more sense to me if you provided such a ratio for mulitple different values of ECS and observed that the higher the ECS the more often a model attends.

Secondly, I am not sure that ratio of attending is computed correctly. As far as I understood for LLama-7B you take hallucinated response (which means that it contradicts external context) and the most attended span in external context. Then you ask gpt-4o to evaluate whether this span supports existence of a conflict in response or not. If that is the case, I do not understand why this experiment shows whether the model attends (the attention span contains part of the context needed for the correct answer) or mis-attends. If attention span supports the existence of a conflict in response it might still not be relevant for the correct response itself, which means a conflict exists but we can not call it a hallucination according to your definition (hallucination = response is contradicting the context or is not supported by it - line 72).

Please correct me if I misunderstood the experiment setting, what is meant by attending, or the way attending and mis-attending is computed.

## Too many hyperparameters
I am afraid that the proposed hallucination detection method is not applicable in practice as it requires a lot of manual hyperparameter tuning. According to the provided values, they all are different per dataset and model (see Appendix I). They include:

- top k % for ECS
- top k % for PKS
- tau threshold for H - page 8 bottom
- alpha and beta for reweighting page 9 top
- chunk size for the chunked version of REDEEP

I suggest that the authors discuss strategies for automating hyperparameter selection or provide guidelines for choosing these parameters in real-world applications.

## Insufficient experiments

### Hallucination detection experiment
- For RagTruth dataset there exist baselines provided by the original paper [2] which perform better than all the baselines considered by you, could you please include them? E.g. Baseline LLama2-13B results fine-tuned on RagTruth have 78.7 F1, see Table 5 in [2] vs yours 78.3 in Table 1. I think the comparison makes a lot of sense since you tune many hyperparams using RagTruth validation dataset while you could simply fine-tune that baseline on the same data instead.
- Same comes for Dolly dataset, please include results for AlignScore and RepC-LE-nn-n2000-e2000 that have 84 and 86 accuracy correspondigly, while the best method provided by you scored 73.73 (LLama2-7B).
- Please also provide results for the Noisy Context split from Dolly [3] dataset because it better approximates realistic RAG application scenario.

### Causal experiment

- First of all, I don’t see how a higher NLL difference for the experimental group than for the control group shows a causal relation between hallucinations occurrence and copying heads neglecting necessary knowledge, could you please elaborate?
- The experiment results are very noisy and it is hard to draw any conclusions from them, for example, boxplot of the experimental group is fully contained within the boxplot of the control group in Figure 5 (b).
- It is not clear how many heads are within experimental and control groups, it can be the case that loss changes are bigger for the experimental group simply because it intervenes in more heads.

### Hallucination generation experiment

Prompt for truthfulness (Appendix L) creates bias, since GPT-4o knows which answer belongs to the baseline and which to AARF. It can influence its answers since usually in scientific papers named methods outperform baselines, which must have been the case on chatgpt training data as well and possibly created such a bias.

Instead, it would be nice to see the results for prompts that contain anonymous names (e.g. model 1 and model 2 instead of baseline and AARF) to avoid the mentioned naming bias and have a randomly shuffled order of AARF and Baseline inputs before showing to GPT-4o to avoid positional bias.

### Lack of sensitivity experiments
Please provide sensitivity experiments to the numerous hyperparameters you introduced (see the section "Too many hyperparameters" for the hyperparameters)

## Unclear writing
- While being core concepts of the paper, Copying Heads (set A) Knowledge FFNs (set F) are not formally defined (line 381). I guess set A is built by taking top-k attention heads after sorting them by ECS while set B is built by taking top-k FFNs after sorting them by PKS, but I could not find it in text.
- Strange ordering equations, for example, Eq. 2 that defines an important part of ECS has an undefined value “a” which is only introduced in Appendix Eq. 8.

## Typos
455: REDEPE

## References

[1] Kortukov, E., Rubinstein, A., Nguyen, E., & Oh, S.J. (2024). Studying Large Language Model Behaviors Under Context-Memory Conflicts With Real Documents.

[2] Wu, Y., Zhu, J., Xu, S., Shum, K., Niu, C., Zhong, R., Song, J., & Zhang, T. (2023). RAGTruth: A Hallucination Corpus for Developing Trustworthy Retrieval-Augmented Language Models. Annual Meeting of the Association for Computational Linguistics.

[3] Hu, X., Ru, D., Qiu, L., Guo, Q., Zhang, T., Xu, Y., Luo, Y., Liu, P., Zhang, Y., & Zhang, Z. (2024). RefChecker: Reference-based Fine-grained Hallucination Checker and Benchmark for Large Language Models. ArXiv, abs/2405.14486.

**Questions:**

- Why LLama2-7B (smaller and older version than others) has better results on Dolly in terms of F1 or Accuracy in Table 1?

**Details Of Ethics Concerns:**

I have no ethics concerns.

---

> ### Author Response · Authors · 2024-11-20
>
> **Q1: No PKS justification: check whether it removes the parametric knowledge bias using some of the setups for that, for example, the one from [1].**
>
> **A1:** Thank you very much for your valuable feedback. Based on your suggestion, we followed the same experimental setup as Table 4 in [1] and used the Llama2 7B model to verify whether removing FFN layers reduces parametric knowledge bias. In our experiments, we removed the Last (32nd) Layer FFN, the 16th Layer FFN, and the 1st Layer FFN, respectively, and observed the parametric knowledge bias as defined in [1]. The experimental results are summarized in the table below:
>
> | Dataset                              | Llama2 7B | Remove Last (32nd) Layer FFN | Remove 16th Layer FFN | Remove 1st Layer FFN |
> |--------------------------------------|-----------|-------------------------------|------------------------|-----------------------|
> | HotpotQA    | +1.44%    | +0.52% (Δ0.92%)              | -1.61% (Δ3.05%)       | +1.33% (Δ0.11%)      |
> | SQuAD     | +5.10%    | +1.24% (Δ3.85%)              | +2.18% (Δ2.92%)       | +3.30% (Δ1.80%)      |
> | NQ    | +4.45%    | +2.55% (Δ1.90%)              | +2.16% (Δ2.29%)       | --                   |
>
>
> In this table, larger values indicate greater parametric knowledge bias. The “/” symbol represents a divide-by-zero error encountered during the calculation process. Considering time constraints, we sampled the first 1,000 examples from the RAG datasets, which is a common setting in RAG scenarios [5].
>
> The experimental results show that removing FFN layers, whether from lower layers or middle-to-upper layers, consistently reduces the parametric knowledge bias. Notably, removing middle-to-upper layer FFNs is more effective in reducing the bias, aligning with our paper’s conclusion that Knowledge FFNs are primarily concentrated in middle-to-upper layers. On the HotpotQA dataset, removing the 16th Layer FFN even resulted in a negative parametric knowledge bias. This further corroborates our conclusion that, in RAG scenarios with accurate and relevant external knowledge, excessive injection of parametric knowledge into the residual stream by Knowledge FFNs within LLMs can lead to hallucinations.
>
> We have added this experiment to our revised paper. Please see Appendix D: JUSTIFICATION FOR PARAMETRIC KNOWLEDGE SCORE (Lines 957–980) in the revised version of our paper.
>
> ---
>
> **Q2: Questionable ECS justification: 1. Purpose of the experiment on Appendix C: DIVE INTO THE RATIONALE BEHIND THE EXTERNAL CONTEXT SCORE; 2. Question about whether ratio of attending is computed correctly.**
>
> **A2:**  First, to address your first question, the purpose of this experiment was not to justify the External Context Score (ECS). For the justification of ECS, we referenced the copying heads theory proposed by Anthropic [2] (as discussed in Lines 297–307 of our paper). Instead, the goal of this experiment was to analyze the reasons behind a low ECS. In the findings presented in Lines 357–362, we identified two potential causes for a low ECS:
> 1. The Copying Heads may occasionally neglect necessary knowledge from the external context.
> 2. The LLM may lose the information retrieved by the Copying Heads during the generation process.
>
> Regarding your second question, we greatly appreciate your suggestion. We recognized the limitations of the previous experiment and added a new experiment with a modified prompt to let GPT-4-o evaluate whether the attended span includes information that should be part of the truthful answer.
> ```
> Prompt: {external context + query}
>
> Truthful Respond: {Truthful Respond}
>
> Given the following context information: {Attend Span}, check if it includes relevant information found in the truthful answer.
>
> Please answer with "Yes" or "No" and provide the reason on a new line.
> ```
> For validation, we used data from RAGTruth where LLaMA2-7B Chat exhibited hallucinations. We extracted truthful answers from other model responses in the RAGTruth dataset for the same questions, filtering out cases where no truthful answer existed and where the PKS was above the average value for the dataset, to minimize the impact of PKS.
>
> | Attention Heads Attend Truthful Information | Attention Heads Mis-Attend Truthful Information |
> |--------------------------------------------|-----------------------------------------------|
> | 65.24%                                     | 34.76%                                        |
>
> As shown in the table, the results indicate that in most cases where hallucinations occur, the attention heads correctly attend to the appropriate external context. This demonstrates that a low ECS is mostly due to the LLM losing the information attended by the attention heads during the generation process.
>
> We have added this experiment to our revised paper. Please see Appendix C: DIVE INTO THE RATIONALE BEHIND THE EXTERNAL CONTEXT SCORE (Lines 932–956) in the revised version of our paper.

---

> > ### Author Response · Authors · 2024-11-20
> >
> > **Q3: Too many hyperparameters.**
> >
> > **A3:** Thank you very much for your valuable feedback. Regarding your concern about hyperparameter tuning in the hallucination detection model ReDeEP, we would like to clarify the following:
> >
> > For **top $K$ for attention heads and FFNs**, **$α$**, **$β$**, and **chunk size for ReDeEP**:
> >
> > 1. **Top $K$ for attention heads and FFNs**:
> >    We use the validation set to perform an initial ranking of attention heads based on their ECS and FFN layers based on their PKS correlations with hallucination labels. Then, we employ a greedy grid search method. For both top-$K$ attention heads and top-$K$ FFN layers, $K$ is searched within the range of [1, 32].
> >
> > 2. **$\alpha$ and $\beta$**:
> >    - The first approach is to treat these as trainable parameters and fit them using regression on the validation set.
> >    - The second approach, which we adopted in our experiments, is grid search. Specifically, we fix one parameter (e.g., $\alpha = 1$) and tune the other ($\beta$). We found that $\beta$ typically falls within the range of (0, 2), and a step size of 0.1 is sufficient for efficient search.
> >
> > 3. **Chunk size**:
> >    For text chunking, we used the `Recursively split by character` method provided by LangChain, which automatically splits text based on characters. Since this is not the focus of our research, we set the maximum chunk size to 256 (a commonly used value in LangChain) and chunk overlap to 20.
> >
> > For **AARF**, the parameters are relatively simpler:
> >
> > 1. **τ**:
> >    A grid search is performed within the range of (0, 1) with a step size of 0.1.
> >
> > 2. **$\alpha_2$**:
> >    We suggest trying grid search with values [1, 2, 5, 10].
> >
> > 3. **$\beta_2$**:
> >    A grid search in the range of (0, 1) with a step size of 0.1 is sufficient.
> >
> > We have added these details to Appendix J: IMPLEMENTATION DETAILS in the revised version of our paper.

---

> ### Author Response · Authors · 2024-11-20
>
> **Q4: Hallucination detection experiment on Baseline LLama2-13B of RAGTruth Paper[3]**
>
> **A4:**
> Thank you for your question. The experimental setup of RAGTruth differs slightly from ours. We focus on hallucination detection for responses from individual models, whereas the RAGTruth paper mixes data from all models for evaluation. Among the baselines provided in the RAGTruth paper, we have already included all except for the fine-tuning of LLaMA2-13B.
>
> However, the full fine-tuning of LLaMA2-13B in the RAGTruth paper was conducted using 4×A100 GPUs, which exceeds the computational resources we have available. Our 4×V100 setup cannot support flash-attention optimizations, resulting in out-of-memory issues during full fine-tuning. We also reached out to the authors of the RAGTruth paper to request testing logs to replicate on our settings, but we did not receive a response. As a result, we conducted LoRA-based fine-tuning instead, whose setting is based on [6]. The results are summarized in the tables below:
>
> | Model      LLaMA2-7b         | RAGTruth AUC | RAGTruth PCC | RAGTruth ACC | RAGTruth Rec | RAGTruth F1 | Dolly (AC) AUC | Dolly (AC) PCC | Dolly (AC) ACC | Dolly (AC) Rec | Dolly (AC) F1 |
> |---------------------|--------------|--------------|--------------|--------------|--------------|----------------|----------------|----------------|----------------|----------------|
> | LLaMA2-7B (LoRA)     | –            | –            | 0.6350       | 0.7078       | 0.6750       | –              | –              | 0.6043         | 0.5918         | 0.6616         |
> | ReDeEP (token)     | 0.7325       | 0.3979       | 0.7067       | 0.6770       | 0.6986       | 0.6884         | 0.3266         | 0.6464         | 0.8070         | 0.7244         |
> | ReDeEP (chunk)     | 0.7458       | 0.4203       | 0.6822       | 0.8097       | 0.7190       | 0.7949         | 0.5136         | 0.7373         | 0.8245         | 0.7833         |
>
> | Model    LLaMA2-13b           | RAGTruth AUC | RAGTruth PCC | RAGTruth ACC | RAGTruth Rec | RAGTruth F1 | Dolly (AC) AUC | Dolly (AC) PCC | Dolly (AC) ACC | Dolly (AC) Rec | Dolly (AC) F1 |
> |---------------------|--------------|--------------|--------------|--------------|--------------|----------------|----------------|----------------|----------------|----------------|
> | LLaMA2-13B (LoRA)    | –            | –            | 0.7034       | 0.6839       | 0.7123       | –              | –              | 0.5545         | 0.6319         | 0.6664         |
> | ReDeEP (token)     | 0.8181       | 0.5478       | 0.7711       | 0.7440       | 0.7494       | 0.7226         | 0.3776         | 0.6465         | 0.8148         | 0.7154         |
> | ReDeEP (chunk)     | 0.8244       | 0.5566       | 0.7889       | 0.7198       | 0.7587       | 0.8420         | 0.5902         | 0.7070         | 0.8518         | 0.7600         |
>
> | Model   LLaMA3-8b            | RAGTruth AUC | RAGTruth PCC | RAGTruth ACC | RAGTruth Rec | RAGTruth F1 | Dolly (AC) AUC | Dolly (AC) PCC | Dolly (AC) ACC | Dolly (AC) Rec | Dolly (AC) F1 |
> |---------------------|--------------|--------------|--------------|--------------|--------------|----------------|----------------|----------------|----------------|----------------|
> | LLaMA3-8B (LoRA)     | –            | –            | 0.6045       | 0.7245       | 0.6613       | –              | –              | 0.5800         | 0.6954         | 0.6495         |
> | ReDeEP (token)     | 0.7522       | 0.4493       | 0.6533       | 0.7984       | 0.7132       | 0.6701         | 0.2421         | 0.6700         | 0.8293         | 0.6901         |
> | ReDeEP (chunk)     | 0.7285       | 0.3964       | 0.6288       | 0.7819       | 0.6947       | 0.7354         | 0.3652         | 0.7100         | 0.8392         | 0.7100         |
>
> The LoRA fine-tuning results could not outperform our model. We acknowledge that full fine-tuning of large-scale LLMs on RAGTruth may yield results comparable to or better than ours. However, full fine-tuning is extremely resource-intensive and tends to overfit the current data distribution, limiting generalization capabilities. This is evident from the weaker performance of the LLaMA2-13B model LoRA-based fine-tuned on RAGTruth when tested on Dolly (AC). In contrast, our model requires no fine-tuning at all.
>
> In deployment scenarios, our model also demonstrates higher efficiency during inference compared to LLM-based methods, further underscoring its practical potential for real-world applications.

---

> ### Author Response · Authors · 2024-11-20
>
> **Q5: Same Baseline comes for Dolly dataset, please include results for AlignScore and RepC-LE-nn-n2000-e2000**
>
> **A5:** The Dolly dataset originates from the RefChecker model paper[4]. AlignScore and RepC-LE-nn-n2000-e2000 are components of RefChecker, and the "84 and 86 accuracy" mentioned for these methods are derived from Table 8 in RefChecker [4]. However, the experiments in Table 8 use 11k human-annotated claim triplets, and while the column is labeled as "Accurate-context," the dataset and task are entirely different from the hallucination detection task in our paper.
>
> For hallucination detection, the relevant results are presented in Table 2 of RefChecker. In our experiments, we compared with RefChecker and ensured fairness by setting both the extractor and checker to `gpt4-o-mini` as described in Appendix I. From Table 1 in our paper, it can be observed that our model outperforms RefChecker, demonstrating the effectiveness of our approach for the hallucination detection task.
>
> ---
>
> **Q6: Please also provide results for the Noisy Context split from Dolly [4] dataset.**
>
> **A6:** Thank you for your suggestion. The Noisy Context split from the Dolly [4] dataset may not be suitable for our scenario. As stated in our paper Line 77-78, our work focuses on detecting RAG hallucinations specifically in cases where the retrieved external context is accurate and relevant.
>
> Our method is designed for scenarios where users upload personal knowledge bases or in realistic RAG application scenarios where filters are applied to remove misinformation and irrelevant information. Such filters are well-studied in fields like misinformation detection and information retrieval, which can largely ensure that the context provided to the RAG model is accurate and relevant.
>
> ---
>
> **Q7: Causal experiment: question about the  NLL difference.**
>
> **A7:** We apologize for any confusion caused. Our intervention experiments were conducted on the RAGTruth dataset, which contains truthful answers. The NLL difference is used to measure the change in the negative log-likelihood of the model generating truthful answers before and after the intervention. If the experimental group shows a larger NLL difference compared to the control group, it indicates that the intervention in the experimental group more significantly disrupts the model's ability to produce truthful answers (i.e., leads to hallucinated answers).
>
> As shown in Figure 5 (Left), the NLL difference is greater for interventions on Copying Heads (experimental group) than for non-Copying Heads (control group). This demonstrates that Copying Heads are more strongly associated with RAG hallucinations compared to non-Copying Heads.
>
> We have added a more detailed explanation of this to the revised paper in Appendix E: DETAILED INTERVENTION PROCEDURES.
>
> ---
>
> **Q8: Causal experiment: question about the noisy results in Figure 5 (b).**
>
> **A8:** Thank you for your question. We acknowledge that Figure 5 (Left) (b), exhibits significant noise and does not effectively demonstrate the impact of FFN interventions. We agree that the intervention experiment you proposed in the "No PKS justification" section better illustrates the causal relationship between FFN layers and RAG hallucinations. Therefore, in the revised version of our paper, we have incorporated the parametric knowledge bias experiment into Section 3.2 EXPERIMENTS (RQ2) to provide stronger evidence that the Knowledge FFNs within LLMs excessively injecting parametric knowledge into the residual stream can lead to RAG hallucinations.
>
> **Q9: How many heads are within experimental and control groups.**
>
> **A9:** In our original paper, we have addressed this concern in detail in the Causal Matching section of Appendix E Detailed Intervention Procedures. To ensure a fair comparison, we matched the top 32 Copying Heads with the nearest non-experimental heads within the same layer. Similarly, we matched the top 5 Knowledge FFNs, identified as being most related to hallucinations, with the nearest FFN modules in adjacent layers.
>
> This Causal Matching approach mitigates the potential bias you mentioned, ensuring that differences in loss changes are not simply due to the experimental group intervening in more heads.
>
> ---

---

> ### Author Response · Authors · 2024-11-20
>
> **Q10: Anonymous prompt for Hallucination generation experiment.**
>
> **A10:**
> Thank you for your suggestion. Following your advice, we conducted the evaluation using anonymous names to mitigate potential naming bias. We ran the evaluation twice: in the first run, AARF and Baseline were anonymized as A and B, respectively; in the second run, their roles were reversed, with AARF and Baseline anonymized as B and A. The final results were obtained by averaging the outcomes of these two evaluations:
>
> | RAGTruth  | AARF Win | Tie | Baseline Win | Dolly (AC)      | AARF Win | Tie | Baseline Win |
> |-----------------|------------|-----|----------|------------------|------------|-----|----------|
> | Llama-2-7B     |  155        | 185 | 110      | Llama-2-7B  | 28         | 54  | 18       |
> | Llama-2-13B    |  160        | 178 | 112      | Llama-2-13B | 30         | 57  | 13       |
> | Llama-3-8B     |  170        | 183 | 97       | Llama-3-8B  | 19         | 66  | 15       |
>
> We observed that compared to the results without anonymization, the number of ties determined by the LLM increased, and the proportion of AARF wins slightly decreased. However, AARF still consistently outperformed the Baseline. This experiment demonstrates that the robustness of AARF's performance persists even under conditions designed to eliminate naming and positional biases.
>
> ---
>
> **Q11: Lack of sensitivity experiments**
>
> **A11:**
> Thank you for your suggestion. We conducted sensitivity analysis experiments on both ReDeEP(Token) and ReDeEP(Chunk) based on the parameter ranges provided in Section J: IMPLEMENTATION DETAILS. The experimental results have been added to the revised paper in Appendix N: SENSITIVITY ANALYSIS.
>
> From the results, we observe a trend of performance either increasing and then stabilizing or decreasing slightly. The overall impact of any single hyperparameter on the model's performance is minor, reflecting the robustness of our approach.
>
> Additionally, we note that the sensitivity curves for the same hyperparameter are similar across different models. For most hyperparameters, the performance extremes appear within the same parameter ranges across models, further demonstrating the cross-model stability of ReDeEP.
>
> Please refer to the revised paper  Appendix N: SENSITIVITY ANALYSIS (Line 1337--1347) for detailed results.
>
> ---
>
> **Q12: Unclear writing: Question about the Copying Heads (set A) Knowledge FFNs (set F)**
>
> **A12:**
> Thank you for your question. Your understanding is correct. We have clarified the details in our revised paper.
>
> ---
>
> **Q13:Strange ordering equations.**
>
> **A13:** We sincerely apologize for the confusion caused by the ordering of equations. Due to the page limit of the main text, we moved the introduction of $a$, which refers to attention weights, to the appendix. This term represents a commonly used concept in the context of our formula, which is why we chose to provide its detailed explanation in Appendix Eq. 8.
>
> ---
>
> **Q14:Why LLama2-7B (smaller and older version than others) has better results on Dolly in terms of F1 or Accuracy in Table 1?**
>
> **A14:**  Thank you for your question. The RAGTruth and Dolly (AC) datasets we used are annotated based on the outputs of different models. Since LLaMA2-7B and LLaMA2-13B may produce different outputs for the same question, the corresponding annotated data also differs. As a result, the metrics are not directly comparable between models. It is possible for a smaller model to outperform a larger model on certain metrics.
>
> This phenomenon is also observed in other hallucination detection papers, such as [6], where in Table 1, LLaMA-7B outperforms LLaMA-13B on the CoQA dataset.

---

> ### Author Response · Authors · 2024-11-20
>
> ---
>
> **Reference:**
>
> [1] Kortukov, E., Rubinstein, A., Nguyen, E., & Oh, S.J. (2024). Studying Large Language Model Behaviors Under Context-Memory Conflicts With Real Documents.
>
> [2] Transformer Circuits Thread, 2021. URL https://transformer-circuits.pub/2021/framework/index.html.
>
> [3] Wu, Y., Zhu, J., Xu, S., Shum, K., Niu, C., Zhong, R., Song, J., & Zhang, T. (2023). RAGTruth: A Hallucination Corpus for Developing Trustworthy Retrieval-Augmented Language Models. Annual Meeting of the Association for Computational Linguistics.
>
> [4] Hu, X., Ru, D., Qiu, L., Guo, Q., Zhang, T., Xu, Y., Luo, Y., Liu, P., Zhang, Y., & Zhang, Z. (2024). RefChecker: Reference-based Fine-grained Hallucination Checker and Benchmark for Large Language Models. ArXiv, abs/2405.14486.
>
> [5] Jin, Jiajie, et al. "FlashRAG: A Modular Toolkit for Efficient Retrieval-Augmented Generation Research." arXiv preprint arXiv:2405.13576 (2024).
>
> [6] Chen, Chao, et al. "INSIDE: LLMs' Internal States Retain the Power of Hallucination Detection." The Twelfth International Conference on Learning Representations.
>
> [7] Yaowei Zheng, Richong Zhang, Junhao Zhang, Yanhan Ye, and Zheyan Luo. 2024. LlamaFactory: Unified Efficient Fine-Tuning of 100+ Language Models. In Proceedings of the 62nd Annual Meeting of the Association for Computational Linguistics.

---

> ### Comment · Reviewer_LdfE · 2024-12-01
> **Thank you + increase score**
>
> Thank you very much for such a detailed response, all my questions are answered! I will increase my score to 8.
>
> I apologize for the late response.

---

> > ### Author Response · Authors · 2024-12-02
> >
> > We are delighted to address your questions and sincerely appreciate your detailed review and positive feedback!

---

### Author Response · Authors · 2024-11-20

We sincerely thank all the reviewers for their thorough evaluations and constructive suggestions, which have significantly contributed to improving our submission. Below, we outline the modifications made in response to the reviewers’ feedback.

1. **Figure 3 (Page 4):** We carefully revised Figure 3 based on Reviewer JEtc’s comments to make it more focused and easier to follow.

2. **Section 5.2 EXPERIMENTS (Page 9):** We conducted anonymous prompt tests for Truthful RAG Generation and added more baseline comparisons.

3. **Appendix C: DIVE INTO THE RATIONALE BEHIND THE EXTERNAL CONTEXT SCORE (Page 17):** We introduced new correlation experiments to validate the rationale behind the external context score.

4. **Appendix D: JUSTIFICATION FOR PARAMETRIC KNOWLEDGE SCORE (Page 18):** We added experiments using parametric knowledge bias to empirically justify the validity of the PARAMETRIC KNOWLEDGE SCORE.

5. **Appendix J: IMPLEMENTATION DETAILS (Page 23):** Additional implementation details for the experiments were included.

6. **Appendix N: SENSITIVITY ANALYSIS (Page 25):** We provided the sensitivity analysis for ReDeEP.

7. **Appendix O: CASE STUDY FOR AARF (Page 26):** We included the case study for AARF to showcase its performance.

Finally, we sincerely thank all the reviewers and the Chairs for their time and efforts in reviewing our work and providing valuable feedback.

---

### Meta-Review · Area_Chair_U4ys · 2024-12-18

**Metareview:**

This paper studies the potential hallucinations issue of Retrieval Augmented Generation (RAG) models in practice and proposes a hallucination detection method, ReDeEP, and a RAG truthfulness improvement method, AARF. Reviewers agreed that the paper is clearly written, and the ideas are well motivated, and experimental results sufficiently demonstrate the effectiveness of the proposed methods. Detecting hallucinations associate with RAG is definitely an important research topic, especially that RAG has been widely applied in many domains nowadays. This paper brings new insights and tools to the field.

**Additional Comments On Reviewer Discussion:**

Reviewers mainly raised some concerns about justifications on technical approaches and insufficient experiments, which have been well addressed by the authors in their rebuttal.

---

### Decision · Program_Chairs · 2025-01-22

Accept (Spotlight)